# GradPruner: Gradient-guided Layer Pruning Enabling Efficient Fine-Tuning and Inference for LLMs

**Wei Huang**[*]  **Anda Cheng**[*]  **Yinggui Wang**[†]
Ant Group
Beijing, China
{hw19970202, andacheng.cad, wyinggui}@gmail.com

## Abstract

Fine-tuning Large Language Models (LLMs) with downstream data is often considered time-consuming and expensive. Structured pruning methods are primarily employed to improve the inference efficiency of pre-trained models. Meanwhile, they often require additional time and memory for training, knowledge distillation, structure search, and other strategies, making efficient model fine-tuning challenging to achieve. To simultaneously enhance the training and inference efficiency of downstream task fine-tuning, we introduce GradPruner, which can prune layers of LLMs guided by gradients in the early stages of fine-tuning. GradPruner uses the cumulative gradients of each parameter during the initial phase of fine-tuning to compute the **I**nitial **G**radient **I**nformation **A**ccumulation Matrix (IGIA-Matrix) to assess the importance of layers and perform pruning. We sparsify the pruned layers based on the IGIA-Matrix and merge them with the remaining layers. Only elements with the same sign are merged to reduce interference from sign variations. We conducted extensive experiments on two LLMs across eight downstream datasets. Including medical, financial, and general benchmark tasks. The results demonstrate that GradPruner has achieved a parameter reduction of 40% with only a 0.99% decrease in accuracy. Our codes are publicly available [1].

## 1 Introduction

LLMs have currently gained remarkable performance across various tasks Grattafiori et al. (2024). However, when handling more specialized tasks, such as medical or financial domains, LLMs often exhibit a decline in performance (Zhang et al., 2023). We can fine-tune them on downstream data to enhance their capabilities. Fine-tuning LLMs on domain-specific data typically requires substantial time and is expensive. For instance, Yang et al (Yang et al., 2024a). trained a medical LLM that took approximately 221 hours. Furthermore, LoRA fine-tuning over pretrained LMs reduces training memory but does not improve inference efficiency (Han et al., 2024).

Structured pruning improves inference efficiency by removing parameter blocks. These structured pruning approaches typically involve two steps Muralidharan et al. (2024); Ma et al. (2023). The first step uses calibration data to identify important parameters within the model. Since this process generally does not introduce additional training, it relies more on the LLM's inherent ability to process the calibration data. However, due to the LLM's suboptimal performance on domain-specific data, this may lead to significant biases. The second step involves training or distilling the pruned model, which requires more time and memory. Some works have also explored structured pruning for efficient LLM training and inference, such as APT Zhao et al. (2024) and SAT Ma et al. (2024). APT can only be applied to LoRA fine-tuning. In SAT, the model structure varies across different training steps, and the final training step restores the model to its dense form, meaning it cannot accelerate inference.

---

[*]Equal contribution. † Corresponding author.
[1]https://github.com/secretflow/ACoLab/tree/main/PaperCode/GradPrune

To ensure the accuracy of downstream tasks while efficiently training and inference with LLMs, we address three key challenges: 1) Developing a method to measure the importance of model parameters tailored to specific downstream data and models without increasing memory or training time, 2) Preserving the original model structure as much as possible while maximizing parameter pruning, and 3) Supporting both full fine-tuning and LoRA fine-tuning. In this paper, we introduce GradPruner, an efficient fine-tuning approach inspired by the observation that loss decreases sharply in the initial fine-tuning steps, indicating rapid learning of downstream tasks. Because different parameters hold varying levels of importance for downstream tasks, this leads to differences in learning capabilities Zhao et al. (2024). Leveraging this insight, to simultaneously save time and memory, we employ LoRA fine-tuning Hu et al. (2022) to compute the accumulation of gradients during the initial phase of training (which is significantly fewer than the total number of training steps) to obtain the Initial Gradient Information Accumulation Matrix (IGIA-Matrix), which is used to assess the importance of each parameter.

We adopted layer-level pruning of the model to address the second and third issues. Through ablation studies, we found that pruning 30% of the layers has almost no impact on the accuracy of downstream tasks, but pruning an additional layer causes a sharp decline in accuracy. To further increase the pruning ratios, we introduced a layer merging method. This method involves sparsifying the pruned layers using the IGIA-Matrix and then merging them with the remaining layers. To minimize interference from sign conflicts, we only merge elements with the same sign. By employing layer merging, we are able to prune three additional layers while maintaining the accuracy of downstream tasks.

We conducted extensive experiments on two LLMs and eight downstream datasets. The results demonstrate that GradPruner can prune 40% of the parameters while ensuring that the accuracy on downstream tasks decreases by only 0.99%. Compared to structured pruning methods, our approach clearly outperforms them. Additionally, the pruned Llama3.1-8B model achieves better accuracy on downstream tasks than the Llama3.2-3B.

This paper makes the following key contributions: 1) We propose GradPruner, which calculates the IGIA-Matrix using the initial gradients from LoRA fine-tuning to evaluate the importance of each model parameter for downstream tasks. 2) To prune as many layers as possible, we sparsify the pruned layers based on the IGIA-Matrix and then merge them with the remaining layers, only combining elements with the same sign. 3) Extensive experiments show that GradPruner has achieved a parameter reduction of 40% with only a 0.99% decrease in downstream tasks accuracy.

## 2 BACKGROUND AND MOTIVATION

### 2.1 PROBLEM FORMULATION

Our goal is to enhance the efficiency of fine-tuning and inference for LLMs while ensuring the accuracy of downstream tasks. Intuitively, the more parameters that are pruned, the faster the training and inference will be, but the accuracy on downstream tasks will decrease. Conversely, the fewer parameters that are pruned, the better the accuracy on downstream tasks, but the training and inference speed will slow down. However, numerous researchers have pointed out that LLMs contain a significant number of redundant parameters Yadav et al. (2023); Muralidharan et al. (2024). Therefore, we aim to identify the minimal set of model parameters that can be retained to preserve the accuracy of downstream tasks.

We define the downstream task dataset as $D$, consisting of $N$ samples $(x_1, y_1), \ldots, (x_N, y_N)$. The objective of our problem is to achieve the maximum pruning ratio $\psi$ while ensuring the minimization of the task loss $L$. We describe the optimization process as:

$$\underset{\theta * \psi}{\arg\min} \frac{1}{|D|} \sum_{x,y \in D} L(x, y|\theta, \psi) \tag{1}$$

where $\theta$ represents all the model parameters.

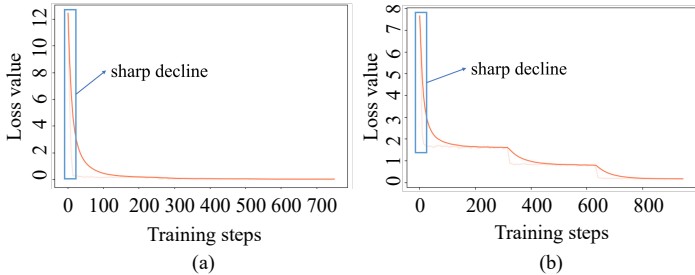

Figure 1: The loss of training while LoRA fine-tuning PubMedQA (a) and PIQA (b) on Llama3.1-8B. The loss value showed a rapid decrease during the initial training steps.

## 2.2 MOTIVATION

Many existing pruning methods rely on calibration data, using forward propagation (such as intermediate layer outputs) to assess the importance of various parameters. Such judgments depend heavily on the LLM's ability to process the calibration data. Nevertheless, when dealing with domain-specific tasks, the LLM's limited processing capabilities may introduce biases.

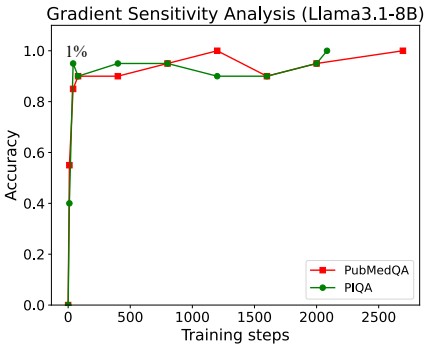

We found that during experiments, when an LLM is fine-tuned on downstream tasks, the loss sharply drops within the initial 1% of the training steps (as shown in Figure 1). This indicates that the model quickly grasps the knowledge required for the downstream task, and different parameters contribute variably to the learning process. This phenomenon provides us with insights.

Based on the above observations and reasoning, it is natural to consider using the initial gradients from LoRA fine-tuning to measure the importance of different parameters. This approach can not only accurately identify parameters that are crucial for downstream tasks but also save time and reduce memory consumption.

Figure 2: Gradient Sensitivity Analysis of the IGIA-Matrix on Llama3.1-8B. We can see that the layer importance measured at such an early stage can accurately reflect the results after the entire training process.

## 2.3 GRADIENT SENSITIVITY ANALYSIS OF THE IGIA-MATRIX

Existing research has demonstrated that performing full training and subsequently using gradient information to assess the importance of different parameters is a reasonable approach Matena & Raffel (2022); Daheim et al. (2023). However, our proposed IGIA-Matrix method requires only the initial 1% of the training steps to measure the importance of each layer. To analyze the relationship between the layer importance derived from our method and that obtained after complete training, we conducted a Gradient Sensitivity Analysis of the IGIA-Matrix.

In our study, we recorded the layer importance at various stages of the training process and used the layer importance obtained after full training as the reference label list. Since we need to prune layers, only the indices of the top 20 most important layers were retained in the label list. We then compared the top 20 layers identified at each stage with this label list. If any layer among the top 20 from a given stage was not present in the label list, we considered that layer's result as mismatched with the labels, resulting in a drop in accuracy. The experimental results of this analysis are presented

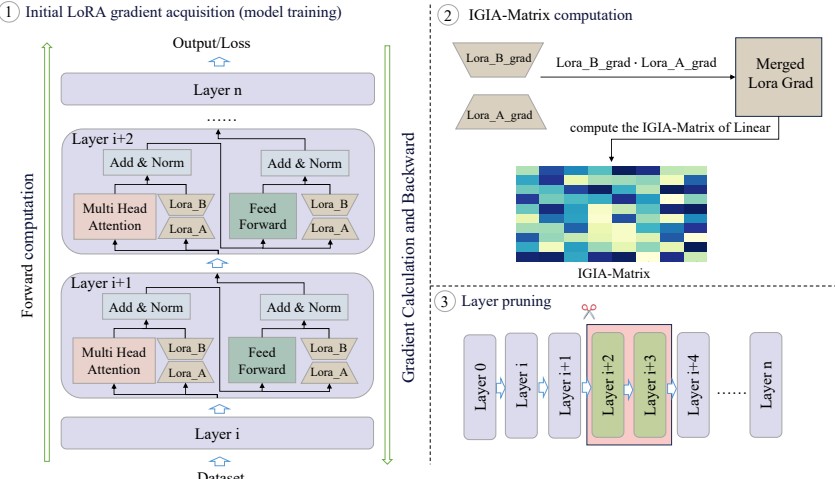

Figure 3: The framework diagram of Parameter Importance Evaluation and Layer Pruning in Grad-Pruner. The first step involves obtaining gradients through a small amount of LoRA fine-tuning. The second step calculates the IGIA-Matrix based on gradients. In the third step, we assess the importance of each parameter and each layer based on the IGIA-Matrix and subsequently prune the layers accordingly.

in Figure 2. From the figure, we can conclude that the layer importance measured at such an early stage can accurately reflect the results after the entire training process.

## 3 METHODOLOGY

In this section, we provide a detailed explanation of GradPruner. GradPruner consists of three steps: **(1) Parameter Importance Measurement Phase**. This step focuses on identifying the model parameters that are important for downstream tasks. **(2) Layer Pruning Phase**. Once the importance of different parameters is determined, the second step assesses the importance of various layers and prunes them accordingly. **(3) Layer Merging Phase**. This step involves merging the pruned layers with the remaining layers.

### 3.1 PARAMETER IMPORTANCE EVALUATION AND LAYER PRUNING

Motivated by the observations from Figure 1, we observe that the loss decreases rapidly during the initial training phase as the model quickly adapts to downstream tasks. Recognizing that different parameters hold varying levels of importance, reflected in their gradient update magnitudes. Based on this insight, we measure parameter significance by capturing gradient values in each step of the early LoRA training phase. The overall process of the algorithm is illustrated in Figure 3.

**Parameter Importance Evaluation:** Formally, we consider a linear layer with weight $W$. The corresponding LoRA weights are $W_A$ and $W_B$. We freeze the parameter $W$ and train the model using the downstream dataset $D$. We define the model to undergo a total of $T$ training steps; however, we only need to obtain the gradients for the first $t$ steps ($t << T$), meaning that the model training terminates after $t$ steps. After $t$ rounds of training, we obtain the per-step gradient values for $W_A$ and $W_B$, denoted as $\nabla_{W_A} L(x, y)$ and $\nabla_{W_B} L(x, y)$, respectively. Specifically, $\nabla_{W_A} L(x, y)$ consists of $t$ gradients: $\{\nabla_{W_A} L(x, y)_1, \nabla_{W_A} L(x, y)_2, ..., \nabla_{W_A} L(x, y)_t\}$. After obtaining $\nabla_{W_A} L(x, y)$ and $\nabla_{W_B} L(x, y)$, we need to evaluate the importance of each parameter in $W$ for the downstream task. First, we align the matrix dimensions of $\nabla_{W_A} L(x, y)$ and $\nabla_{W_B} L(x, y)$ with those of $W$. We Inspire for LoRA to be mergeable with the original parameters after fine-tuning, so we align them with $W$ by performing matrix multiplication on $\nabla_{W_A} L(x, y)$ and $\nabla_{W_B} L(x, y)$ to simulate the gradient of $W$. The formula is as follows:

$$\nabla_W L(x, y)_i \overset{\text{sim}}{=} \nabla_{W_B} L(x, y)_i \cdot \nabla_{W_B} L(x, y)_i, \quad i \in \{1, ..., t\} \tag{2}$$

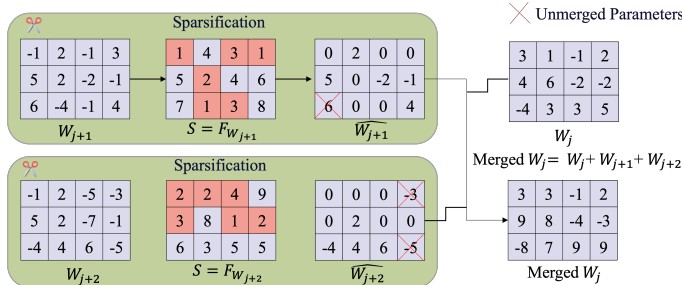

Figure 4: The framework diagram of Layer Merging in GradPruner. The first step is to sparsify the pruned modules using the IGIA-Matrix as the criterion. The second step is to merge the pruned layer with the preceding retained layer based on their signs.

Where, $sim$ represents the simulated gradient.

Using Equation (2), we have obtained the gradients of the $W$ at each step during the first $t$ training steps. To comprehensively evaluate the relationship between the gradients from the initial $t$ training steps and parameter importance, we calculate the IGIA-Matrix using the following formula.

$$F_W = \frac{1}{t} \sum_{i=1}^{t} (\nabla_W L(x, y)_i)^2 \tag{3}$$

We obtain the IGIA-Matrix $F_W$ for $W$ through the steps above. Similarly, all linear layers in LLM can acquire their corresponding IGIA-Matrix using this approach.

**Layer Pruning:** We choose to prune the model's layers based on the following two considerations: 1) Preservation of the Model's Overall Structure: To ensure that the overall architecture of the model remains as unchanged as possible. 2) Impact on Downstream Task Accuracy: Our experiments demonstrate that pruning the parameters of important layers adversely affects the accuracy of downstream tasks (see ablation study). We sum the IGIA-Matrix of all linear layers within each layer of the LLM to obtain the importance score of the current layer, as shown in the following:

$$Layer_j = \sum_{k=1}^{M} \sum_{l=1}^{H} F_{W_{kl}} \tag{4}$$

Where $Layer_j$ represents the importance score of the $j$-th layer, $M$ denotes the number of linear layers within the $j$-th layer, and $H$ signifies the number of parameters in each linear layer. Based on the above formula, we can obtain the importance score for each layer in the model. The pseudocode for layer pruning can be found in Appendix A.

## 3.2    LAYER MERGING

When only performing layer pruning, our ablation study indicates that pruning more than 30% of the layers results in a significant decrease in accuracy. To prune as many layers of the model as possible, we do not directly discard the pruned layers but instead merge them with the remaining layers. The overall process of the algorithm is illustrated in Figure 4.

To merge multiple linear layers $\{W_j\}_{j=1}^{n}$ from different layers. We designate $W_1$ as the linear layer to be retained and $\{W_2, \ldots, W_n\}$ as the linear layers to be pruned. Our method follows two steps to perform the merging.

1) **Sparsification**: Sparsification aims to reduce interference between pruned and retained layers while preserving downstream task accuracy. Research shows that keeping only a subset of model parameters can maintain this accuracy. The key challenge is determining which parameters are most important. To tackle this, we use the IGIA-Matri as a metric to evaluate the importance of each parameter for downstream tasks. The IGIA-Matrix corresponding to $\{W_2, \ldots, W_n\}$ are $\{F_{W_2}, \ldots, F_{W_n}\}$. We retain the top-$p\%$ of parameters based on the magnitude of the IGIA-Matrix

| Method | PubMedQA | MedMCQA | BillSum | FinGPT | HellaSwag | WinoGrande | ARC | PIQA | Avg. |
|---|---|---|---|---|---|---|---|---|---|
| | | | *Llama3.1-8B* | | | | | | |
| | | | *Full Fine-Tuning (FFT)* | | | | | | |
| Dense Model | 0.593 | 0.572 | 0.696 | 0.869 | 0.943 | 0.868 | 0.865 | 0.867 | 0.784 |
| LLMPruner | 0.560 | 0.521 | 0.641 | 0.818 | 0.898 | 0.810 | 0.817 | 0.805 | 0.734 |
| Laco | 0.556 | 0.514 | 0.649 | 0.814 | 0.901 | 0.818 | 0.824 | 0.809 | 0.736 |
| MINITRON | 0.555 | 0.527 | 0.640 | 0.808 | 0.898 | 0.814 | 0.826 | 0.803 | 0.734 |
| SAT | 0.567 | 0.552 | 0.656 | 0.832 | 0.908 | 0.835 | 0.822 | 0.833 | 0.750 |
| FT(Llama3.2) | 0.591 | 0.568 | 0.688 | 0.867 | 0.932 | 0.836 | 0.874 | 0.861 | 0.777 |
| GradPruner(ours) | 0.591 | 0.586 | 0.687 | 0.867 | 0.939 | 0.861 | 0.849 | 0.876 | **0.782** |
| | | | *Parameter-Efficient Fine Tuning (LoRA)* | | | | | | |
| Dense Model | 0.607 | 0.633 | 0.677 | 0.831 | 0.959 | 0.821 | 0.931 | 0.893 | 0.794 |
| LLMPruner | 0.538 | 0.555 | 0.636 | 0.766 | 0.883 | 0.790 | 0.870 | 0.826 | 0.733 |
| Laco | 0.553 | 0.556 | 0.631 | 0.775 | 0.907 | 0.792 | 0.875 | 0.837 | 0.740 |
| MINITRON | 0.546 | 0.557 | 0.631 | 0.765 | 0.899 | 0.781 | 0.873 | 0.822 | 0.734 |
| SAT | 0.550 | 0.596 | 0.621 | 0.779 | 0.909 | 0.780 | 0.882 | 0.844 | 0.745 |
| APT | 0.561 | 0.607 | 0.645 | 0.802 | 0.922 | 0.790 | 0.891 | 0.859 | 0.759 |
| FT(Llama3.2) | 0.592 | 0.619 | 0.662 | 0.825 | 0.926 | 0.808 | 0.902 | 0.859 | 0.774 |
| GradPruner(ours) | 0.594 | 0.637 | 0.659 | 0.817 | 0.954 | 0.812 | 0.923 | 0.891 | **0.786** |
| | | | *Mistral-7B* | | | | | | |
| | | | *Full Fine-Tuning (FFT)* | | | | | | |
| Dense Model | 0.591 | 0.583 | 0.684 | 0.862 | 0.841 | 0.878 | 0.905 | 0.903 | 0.781 |
| LLMPruner | 0.547 | 0.512 | 0.639 | 0.817 | 0.810 | 0.815 | 0.867 | 0.839 | 0.730 |
| Laco | 0.552 | 0.525 | 0.641 | 0.819 | 0.825 | 0.828 | 0.864 | 0.857 | 0.738 |
| MINITRON | 0.544 | 0.511 | 0.650 | 0.823 | 0.822 | 0.830 | 0.851 | 0.846 | 0.734 |
| SAT | 0.561 | 0.543 | 0.657 | 0.823 | 0.834 | 0.836 | 0.867 | 0.870 | 0.748 |
| GradPruner(ours) | 0.586 | 0.568 | 0.670 | 0.846 | 0.840 | 0.860 | 0.895 | 0.897 | **0.770** |
| | | | *Parameter-Efficient Fine Tuning (LoRA)* | | | | | | |
| Dense Model | 0.607 | 0.565 | 0.681 | 0.853 | 0.963 | 0.846 | 0.909 | 0.896 | 0.790 |
| LLMPruner | 0.527 | 0.508 | 0.625 | 0.793 | 0.904 | 0.801 | 0.840 | 0.827 | 0.728 |
| Laco | 0.541 | 0.499 | 0.643 | 0.804 | 0.904 | 0.812 | 0.851 | 0.846 | 0.737 |
| MINITRON | 0.533 | 0.504 | 0.633 | 0.802 | 0.900 | 0.805 | 0.840 | 0.835 | 0.731 |
| SAT | 0.545 | 0.524 | 0.629 | 0.807 | 0.931 | 0.815 | 0.857 | 0.843 | 0.743 |
| APT | 0.555 | 0.533 | 0.631 | 0.813 | 0.926 | 0.815 | 0.866 | 0.854 | 0.750 |
| GradPruner(ours) | 0.588 | 0.565 | 0.659 | 0.840 | 0.963 | 0.832 | 0.893 | 0.896 | **0.780** |

Table 1: The main results of our experiments under 40% sparsity pruning. "Avg." refers to the average score between eight datasets. "Dense Model" represents the results of the unpruned LLMs after fine-tuning. "FT" represents fine-tuning Llama3.2-3B. Since APT can only be applied to LoRA fine-tuning, we only report the results of APT in the context of LoRA fine-tuning.

and set the remaining parameters to zero, thereby creating $\{\hat{W}_2, \ldots, \hat{W}_n\}$. Notably, we do not need to sparsify $W_1$, as it is a more critical linear layer, and sparsifying it could adversely affect the accuracy of downstream tasks.

2) **Symbol-based merging**. A given parameter may have positive values for some layers and negative values for others. In both cases, simply merging these values can lead to interference, thereby shrinking the value of that parameter in the merged layer. Therefore, we using the parameter signs of $W_1$ as the total sign. Only the parameters in $\{\hat{W}_2, \ldots, \hat{W}_n\}$ with signs matching the total sign are merged with $W_1$. For more details, please refer to Equation 5.

$$M\left(W_j\right)_{kl} = \begin{cases} (W_j)_{kl} & if\ (\gamma_j)_{kl}! = (\gamma_{j+1})_{kl}! = (\gamma_{j+n})_{kl} \\ (W_j)_{kl} + (W_{\hat{j+1}})_{kl} & if\ (\gamma_j)_{kl} == (\gamma_{j+1})_{kl}! = (\gamma_{j+n})_{kl} \\ (W_j)_{kl} + (W_{\hat{j+n}})_{kl} & if\ (\gamma_j)_{kl}! = (\gamma_{j+1})_{kl} == (\gamma_{j+n})_{kl} \end{cases} \quad (5)$$

Where $(W_j)_{kl}$ denotes the $l$-th parameter of the $k$-th linear layer in the $j$-th layer, and $\gamma$ represents the sign matrix corresponding to the parameter matrix of the linear layer. Similarly, each linear layer in the pruned layer can be merged with the corresponding linear layer of the retained layer following the above process. The pruned layer is only merged with the preceding retained layer.

## 4 EXPERIMENTAL SETTINGS

### 4.1 DATASETS AND SETTING

To comprehensively evaluate the effectiveness of GradPruner in the vertical domain, we conducted extensive experiments using two widely adopted LLMs and eight downstream datasets. For the

| Method | Train Time($\Downarrow$) | Train Mem($\Downarrow$) | Inf Time($\Downarrow$) | Inf Mem($\Downarrow$) |
|---|---|---|---|---|
| *Full Fine-Tuning (FFT)* | | | | |
| Dense Model | 100.0% | 100.0% | 100.0% | 100.0% |
| SAT | 75.5% | 79.4% | 98.9% | 103.6% |
| Laco | 73.8% | 64.4% | 59.7% | 61.3% |
| LLMPruner | 78.3% | 284.4% | 67.4% | 65.3% |
| GradPruner | 62.4% | 65.8% | 61.5% | 60.9% |
| *Parameter-Efficient Fine Tuning (LoRA)* | | | | |
| Dense Model | 100.0% | 100.0% | 100.0% | 100.0% |
| APT | 158.0% | 65.9% | 87.5% | 62.4% |
| SAT | 74.8% | 82.9% | 102.3% | 99.4% |
| Laco | 71.5% | 65.7% | 61.1% | 61.4% |
| LLMPruner | 81.8% | 266.4% | 69.7% | 64.6% |
| GradPruner | 66.9% | 64.4% | 61.8% | 60.7% |

Table 2: Comparison of GradPruner with other methods in terms of training and inference time, as well as GPU memory usage. Our measurement method is based on APT's paper. All efficiency metrics are normalized to Dense Model, that is, the relative time or memory overhead compared to the Dense Model. $\Downarrow$ denotes smaller is better. We do not compare to distillation method (MINITRON) because the training cost of distillation is too large.

LLMs, we selected Llama3.1-8B Dubey et al. (2024) and Mistral-7B-v0.3 Jiang et al. (2023). Regarding the downstream datasets, we included four specialized domain datasets: PubMedQA Jin et al. (2019) and MedMCQA Pal et al. (2022), which focus on medical tasks, and BillSum Kornilova & Eidelman (2019) and fingpt-sentiment-train (FinGPT) Yang et al. (2023), which pertain to financial tasks. Additionally, we incorporated four general-domain reasoning benchmark datasets: HellaSwag, WinoGrande, ARC, and PIQA Yao et al. (2024).

Due to space constraints in the main paper, a detailed description of the datasets and the setup of our experiment is presented in the appendix C.

Our evaluation metrics are formulated based on the characteristics of the dataset. The QA data in the medical and financial fields, we adopt the method of evaluating the similarity between the output from LLMs and the standard label. We evaluate using 1/2 * ( BertScore (Zhang* et al., 2020) + ROUGE-L (Lin, 2004)). As for other reasoning benchmarks, we directly calculate the Accuracy score.

## 4.2 BASELINES

We conducted extensive comparisons between GradPruner and six baselines. Categorizing the baselines into three groups. The first group consists of structured pruning methods focused on addressing how to efficiently train and infer when LLMs are adapted to downstream tasks. 1) APT Zhao et al. (2024), which dynamically adds salient tuning parameters for fast and accurate convergence while discarding unimportant parameters to improve efficiency; 2) SAT Ma et al. (2024), which extends existing neuron importance evaluation metrics and introduces a ladder omission rate scheduler.

The second group of structured pruning methods focuses on accelerating inference. 1) LLM-Pruner Ma et al. (2023) adopts structural pruning that selectively removes non-critical coupled structures based on gradient information. 2) LaCo Yang et al. (2024b), a concise layer-wise structured pruner called Layer Collapse, in which rear model layers collapse into a prior layer; 3) MINITRON Muralidharan et al. (2024), which combines depth, width, attention, and MLP pruning with knowledge distillation-based retraining. We fine-tuned the models pruned by LLMPruner, LaCo, and MINITRON on downstream data for comparison. Since MINITRON requires pre-training data for knowledge distillation, which we could not access, we used the Alpaca dataset as a substitute.

The third group of baselines consists of smaller parameter models. We used the fine-tuning of Llama3.2-3B Dubey et al. (2024) on downstream tasks as a comparison method.

It is noted that, to ensure a fair comparison, the pruning baselines also utilize the downstream datasets as calibration data.

| Number | Method | PubMedQA | MedMCQA | BillSum | FinGPT | HellaSwag | WinoGrande | ARC | PIQA | Avg. |
|--------|--------|----------|---------|---------|--------|-----------|------------|-----|------|------|
| | | | | *Llama3.1-8B* | | | | | | |
| | | | | *Full Fine-Tuning (FFT)* | | | | | | |
| | Dense Model | 0.593 | 0.572 | 0.696 | 0.869 | 0.943 | 0.868 | 0.865 | 0.867 | 0.784 |
| 3 | GradPruner | 0.591 | 0.586 | 0.687 | 0.867 | 0.939 | 0.861 | 0.849 | 0.876 | 0.782 |
| | w/o Merging | 0.560 | 0.535 | 0.663 | 0.816 | 0.893 | 0.803 | 0.830 | 0.826 | 0.741 |
| 2 | GradPruner | 0.593 | 0.590 | 0.695 | 0.867 | 0.942 | 0.868 | 0.871 | 0.861 | 0.786 |
| | w/o Merging | 0.581 | 0.566 | 0.677 | 0.841 | 0.930 | 0.840 | 0.846 | 0.843 | 0.767 |
| 1 | GradPruner | 0.590 | 0.588 | 0.695 | 0.866 | 0.945 | 0.863 | 0.866 | 0.868 | 0.785 |
| | w/o Merging | 0.585 | 0.580 | 0.688 | 0.855 | 0.936 | 0.852 | 0.853 | 0.851 | 0.775 |

Table 3: Experimental results on the impact of the number of merged layers on downstream task accuracy. w/o Merging indicates that no layer merging. The experimental results of LoRA fine-tuning and Mistral-7B are presented in the appendix D.

# 5 MAIN RESULTS

Table 1 evaluates different pruning methods and fine-tuning approaches across multiple datasets. The results show that GradPruner consistently outperforms other methods, achieving the best performance on all datasets. To summarize the findings, we compute the average scores of each pruner, revealing that GradPruner incurs only a 0.99% average performance drop comparing with the dense model. Notably, our pruned models outperform directly fine-tuned versions of Llama3.2-3B, demonstrating competitiveness with advanced pre-trained LLMs. GradPruner exhibits particularly strong gains, improving accuracy by approximately 5 percentage points on average for LLMPruner, Laco, and MINITRON, while also surpassing SAT and APT in accuracy.

In addition to accuracy, Table 2 compares GradPruner with other methods in terms of time and GPU memory usage during training and inference. From Table 2, GradPruner achieves approximately 36% reductions in both training time and memory usage, along with 39% savings in inference time and memory, performing comparably to Laco. In contrast, the APT method shows significantly higher training time due to its reliance on knowledge distillation, while the SAT method exhibits similar inference time and memory usage to the dense model, as it gradually restores pruned parameters during training.

# 6 ABLATION STUDY AND ANALYSIS

We conducted five ablation experiments to validate the design decisions and effectiveness of the proposed method.

**Perform Only the Layer Pruning in GradPruner:** Figure 5 presents the results of using only the layer pruning strategy. For Llama3.1-8B, pruning up to 10 layers has minimal impact on downstream task accuracy. However, beyond 10 layers, accuracy begins to decline significantly, as increasingly critical layers are removed. These findings suggest that while layer pruning alone is effective to a certain extent, it becomes insufficient for aggressive pruning. The experimental results of Mistral-7B can be found in the appendix D.

**The Impact of the Number of Merged Layers on Accuracy:** Table 3 demonstrates the impact on downstream task accuracy after incorporating the layer merging algorithm on top of layer pruning. From the table, we can observe that for both models, whether fine-tuned using FFT or LoRA, layer merging significantly improves accuracy. Specifically, for Llama3.1-8B, merging 1 to 3 layers on top of pruning 10 layers results in a substantial accuracy improvement, nearly matching the dense model accuracy. Overall, layer merging ensures the accuracy of downstream tasks while further increasing the pruning rate.

**The Impact of Different Pruning Ratios:** To investigate the impact of different sparsity rates on downstream task accuracy, we conducted relevant experiments. Specifically, we set the sparsity rates to 50%, 60%, 70%, 80%, and 90% to observe their effects on task performance. The experimental results are shown in Figure 6. By analyzing the data in the figure, we found that both excessively high and low sparsity rates have varying degrees of impact on task accuracy. When the sparsity rate

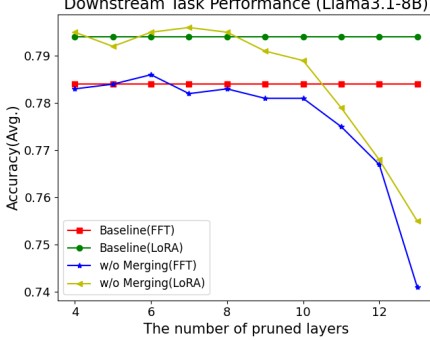

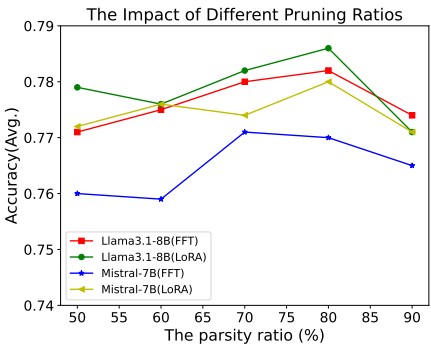

Figure 5: Experimental results of performing Only the layer pruning in GradPruner. We report the average accuracy across eight datasets.

Figure 6: Experimental results of different pruning ratios in GradPruner. We report the average accuracy across eight datasets.

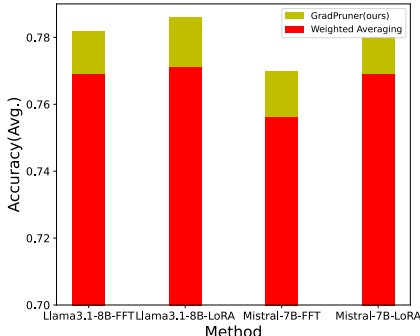

Figure 7: Experimental results of replacing symbol-based merging with weighted averaging in GradPruner. We report the average accuracy across eight datasets.

is too high, too many important parameters are pruned; conversely, when the sparsity rate is too low, too many redundant parameters are retained, leading to increased interference among parameters during layer merging. Based on the ablation study results, we selected an 80% sparsity rate as the optimal setting.

**Replacing Symbol-Based Merging with Weighted Averaging:** GradPruner, during layer merging, employs a sign-based merging strategy on top of sparsification. To demonstrate its superiority, we replaced the sign-based merging with a weight-averaging method for comparison. The results are shown in Figure 7.

It is evident that sign-based merging significantly outperforms weight averaging. This indicates that merging only parameters with matching signs can effectively reduce interference among different parameters, thereby enhancing the accuracy of downstream tasks.

**Replacing Layer Pruning with Kernel Pruning:** Readers may wonder why GradPruner focuses on pruning layers rather than adopting a finer-grained approach, such as pruning kernels (rows or columns). Due to space constraints in the main paper, we have included this portion of the experimental results in the appendix D.

## 7 RELATED WORK

Structured Model Pruning refers to techniques for improving model efficiency by sparsification or parameter removal. While some considerations for pruning are from the perspective of hardware (Xia et al., 2023). Generally, considerations are made from the depth or width of the LLMs (Ling et al., 2024). During the pruning process, the parameters of the target part are typically directly deleted without considering the reliability of the parameter-judgment process (Men et al., 2025), and the performance of the model often suffers significant losses. As a result, some works consider merging parameters into more important layers (Liu et al., 2024; Yang et al., 2024b). However, these works always consider pruning in depth and width respectively. NASH constructs a narrow encoder and a shallow decoder for T5 models (Ko et al., 2023). He et al. (2025) investigate

redundancy across different modules within Transformers, including Layers, MLP, and Attention layers, none of them consider jointly pruning by a combination of multiple importance scores.

Sparse Training is a training approach that makes model parameters sparsely distributed during training (Graesser et al., 2022). Frankle & Carbin (2019); Evci et al. (2019) discovered that training sparse networks from a random initialization is difficult compared to dense neural networks. Despite this, dynamic sparse training (Liu et al., 2021), and one-shot pruning (Tanaka et al., 2020) were proposed to improve sparse training. Sparse Training needs to change the structure of the model during initialization, so it is hard to adapt to downstream tasks for well-trained LLMs.

## 8 CONCLUSION

In this study, we propose GradPruner to address the challenges of time-consuming fine-tuning and high GPU memory usage in LLMs. We compute the initial training gradients to obtain the IGIA-Matrix, which is used to evaluate the importance of different layers. To further increase the number of pruned layers, we introduce a layer merging technique, which includes sparsification using the IGIA-Matrix and resolving sign interference issues. We conducted experiments on various downstream datasets and models. The results demonstrate that GradPruner can maintain nearly the same accuracy while reducing the number of parameters by 40%.

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

## A   PSEUDOCODE FOR LAYER PRUNING

---

**Algorithm 1** Layer Pruning

---

**Input**: Dataset $D$, total training steps $T$, initial training steps $t$ (with $t << T$), model $M$, a linear layer $W$ within $M$, the corresponding LoRA weights $W_A$ and $W_B$ for $W$, the number of pruned layers $N$, and the total number of layers $K$

**Output**: Pruned Model $M_{prune}$

1: Initialize the weights of $W_A$ and $W_B$.
2: ▷ Step 1: Obtain the gradient corresponding to $W$ at every training step during training.
3: **for** i in $\{1,...,t\}$ **do**
4:      $\nabla_{W_A} L(x,y)_i, \nabla_{W_B} L(x,y)_i \leftarrow$ Training-and-obtaining-gradients($L(x,y)_i$)  #  Obtaining Gradients of $W_A$ and $W_B$
5:      $\nabla_W L(x,y)_i = \nabla_{W_B} L(x,y)_i \cdot \nabla_{W_B} L(x,y)_i$
6: **end for**
7: ▷ Step 2: Calculate the IGIA-Matrix corresponding to $W$.
8: $F_W = \frac{1}{t} \sum_{i=1}^{t} (\nabla_W L(x,y)_i)^2$
9: ▷ Step 3: Obtain each layer's importance score based on the IGIA-Matrix.
10: **for** j in 1,...,K **do**
11:      $Layer_j = \sum_{k=1}^{M} \sum_{l=1}^{H} F_{W_{kl}}$
12: **end for**
13: ▷ Step 4: Prune model M based on the layer importance scores.
14: $M_{prune} \leftarrow$ Prune($M$, $Layer_j$, $N$) # Pruned layer
15: **return** $M_{prune}$

---

## B   THEORETICAL ANALYSIS

### B.1   THEORY OF EARLY GRADIENT ACCUMULATION

During the early stages of fine-tuning, the model rapidly shift from its pre-trained state toward a configuration better suited for the downstream task. The accumulated gradient information during this phase capture the most informative weight updates, reflecting the critical sub-networks that contribute significantly to task-specific learning. This phenomenon resonates with the foundation of the Lottery Ticket Hypothesis (LTH; Frankle & Carbin, 2019), which posits that a randomly initialized, dense neural network contains a subnetwork—referred to as a "winning ticket"—that, when trained in isolation, can achieve performance comparable to the original network. LTH-style pruning typically involves training the network for a few steps and then identifying important weights based on their magnitudes. Notably, our proposed method can be interpreted as a gradient-based generalization of this idea. The magnitude of accumulated gradient information over initial fine-tuning steps serves as a proxy for parameter sensitivity to task-specific signals. This information aligns closely with the most salient features in the loss landscape, and pruning along the less sensitive directions preserves the core learning capacity of the model. Therefore, our gradient-based early pruning method is consistent with LTH's principle of identifying winning subnetworks through short periods of training. This connection supports the effectiveness of our approach in capturing long-term parameter importance through early gradient accumulation.

### B.2   THEORY OF MERGING IN GRADPRUNER

In principle, GradPruner is theoretically equal to the isotropic merging method with adaptive importance weights. Furthermore, this adaptive importance weighting scheme can also be similarly applied to Fisher merging. We will elaborate on these two parts in detail below.

**Isotropic merging with adaptive importance weights.**   To merge $M$ layers, isotropic merging with per-layer weights (Matena & Raffel, 2022) approximates the posterior distribution of each layer using an isotropic Gaussian whose mean is set to the layer's parameters. This approach introduces

layer-specific scalar hyperparameters $\lambda_i$, for $i \in \{1, \ldots, M\}$, which can be formally expressed as

$$\theta^* = \arg \max_\theta \sum_{i=1}^M \lambda_i \log p(\theta \mid \theta_i, I),$$

where $p(\theta \mid \theta_i, I)$ denotes the probability density function of the isotropic Gaussian posterior, and the hyperparameters satisfy $\lambda_i \geq 0$ and $\sum_{i=1}^M \lambda_i = 1$. These hyperparameters govern the relative importance assigned to each layer during the merging process. For instance, if all layers are assumed to contribute equally, one may set $\lambda_i = \frac{1}{M}$ for all $i$. However, this approach suffers from two primary limitations: (1) How should the importance weights be adaptively determined when the layers exhibit differing levels of importance? (2) When individual weights within the same layer possess varying degrees of importance, how can adaptive weight-specific coefficients be assigned?

GradPruner introduces an adaptive weighting scheme to solve the above challenges. Specifically, consider two layers to be merged: let $W_r$ denote the base layer to be retained, and $W_m$ the layer to be merged. To assign appropriate weights $\lambda_r^j$ and $\lambda_m^j$ to the $j$-th parameter entries $W_r^j$ and $W_m^j$, GradPruner can be formulated as the following adaptive weight assignment function:

$$(\lambda_r^j, \lambda_m^j) = \begin{cases} (0.5, 0.5), & \text{if} \quad (F_{W_m})_j^2 \geq T \quad \text{and} \quad \text{Sign}(W_m^j) = \text{Sign}(W_r^j), \\ (1.0, 0.0), & \text{otherwise.} \end{cases} \tag{6}$$

where $T$ denotes a pruning threshold. This formulation implies that if the squared gradient information of a weight in the merged layer exceeds the threshold $T$ and its sign aligns with that of the corresponding weight in the retained layer, both weights are deemed equally important and are assigned equal coefficients. Otherwise, the weight from the merged layer is considered negligible, and only the retained weight is preserved. GradPruner's importance criterion simultaneously leverages (1) magnitude — quantified by the squared gradient information — and (2) directional consistency — captured by the agreement in sign between the two weights — as complementary indicators of parameter significance. This design integrates insights from prior work: methods such as Matena & Raffel (2022); Daheim et al. (2023) emphasize the utility of gradient magnitude in assessing parameter importance, while approaches like Kim et al. (2025); Yadav et al. (2023) demonstrate the efficacy of sign consistency as a relevance signal. GradPruner thus unifies both perspectives into a single adaptive framework. Consequently, the merged weight under isotropic merging with adaptive importance weights is given by $\theta^j = \lambda_r^j \theta_r^j + \lambda_m^j \theta_m^j$.

**Integrating GradPruner's Adaptive Weights with Fisher Merging.** As outlined above, Grad-Pruner fundamentally constitutes an adaptive strategy to determine importance weights. In this work, we apply this strategy to isotropic merging. Moreover, we demonstrate that GradPruner's adaptive weighting mechanism can be seamlessly extended to Fisher Merging (FM; Matena & Raffel, 2022). Similar to isotropic merging, FM also formulates the merging objective as $\theta^* = \arg \max_\theta \sum_{i=1}^M \lambda_i \log p(\theta \mid \theta_i, I)$, but differs in that it employs the Laplace approximation to the posterior $p$, yielding a Gaussian approximation $\mathcal{N}(\theta, H^{-1})$, where $H$ is the Hessian matrix. To render computation tractable, FM approximates the Hessian using the diagonal of the Fisher information matrix $F$. The resulting closed-form expression for the merged parameter is $\theta^j = \frac{\sum_{i=1}^M \lambda_i F_i^j \theta_i^j}{\sum_{i=1}^M \lambda_i F_i^j}$. When integrating GradPruner with FM, the adaptive weighting scheme modifies only the assignment of $\lambda_i$, leaving the computation of the Fisher matrix unchanged. Therefore, in the case of merging two layers, the final merged weight becomes $\theta^j = \frac{\lambda_r^j F_r^j \theta_r^j + \lambda_m^j F_m^j \theta_m^j}{\lambda_r^j F_r^j + \lambda_m^j F_m^j}$, where $(\lambda_r^j, \lambda_m^j)$ are adaptively determined via GradPruner as described above.

## C SUPPLEMENTARY EXPERIMENTAL SETUP

### C.1 DATASET DESCRIPTION

1) PubMedQA (Jin et al., 2019) consists of 19717 scientific publications from the PubMed database of diabetes classified into one of three classes, which is in the medical domain.

2) MedMCQA Pal et al. (2022) is a large-scale, Question Answering (QA) dataset designed to address real-world medical questions. We randomly selected 40,000 samples as the training set.

| Number | Method | PubMedQA | MedMCQA | BillSum | FinGPT | HellaSwag | WinoGrande | ARC | PIQA | Avg. |
|---|---|---|---|---|---|---|---|---|---|---|
| | | | | | *Llama3.1-8B* | | | | | |
| | | | | *Parameter-Efficient Fine Tuning (LoRA)* | | | | | | |
| | Dense Model | 0.607 | 0.633 | 0.677 | 0.831 | 0.959 | 0.821 | 0.931 | 0.893 | 0.794 |
| 3 | GradPruner | 0.594 | 0.637 | 0.659 | 0.817 | 0.954 | 0.812 | 0.923 | 0.891 | 0.786 |
| | w/o Merging | 0.576 | 0.599 | 0.646 | 0.787 | 0.902 | 0.776 | 0.885 | 0.857 | 0.755 |
| 2 | GradPruner | 0.597 | 0.637 | 0.673 | 0.818 | 0.958 | 0.818 | 0.928 | 0.886 | 0.789 |
| | w/o Merging | 0.587 | 0.601 | 0.656 | 0.808 | 0.931 | 0.802 | 0.899 | 0.858 | 0.768 |
| 1 | GradPruner | 0.605 | 0.632 | 0.675 | 0.824 | 0.958 | 0.825 | 0.933 | 0.885 | 0.792 |
| | w/o Merging | 0.593 | 0.616 | 0.665 | 0.816 | 0.944 | 0.811 | 0.919 | 0.877 | 0.779 |
| | | | | | *Mistral-7B* | | | | | |
| | | | | *Full Fine-Tuning (FFT)* | | | | | | |
| | Dense Model | 0.591 | 0.583 | 0.684 | 0.862 | 0.878 | 0.905 | 0.903 | 0.781 | |
| 2 | GradPruner | 0.586 | 0.568 | 0.670 | 0.846 | 0.840 | 0.860 | 0.895 | 0.897 | 0.770 |
| | w/o Merging | 0.551 | 0.568 | 0.646 | 0.822 | 0.811 | 0.805 | 0.856 | 0.863 | 0.740 |
| 1 | GradPruner | 0.585 | 0.578 | 0.685 | 0.860 | 0.840 | 0.872 | 0.902 | 0.897 | 0.777 |
| | w/o Merging | 0.579 | 0.567 | 0.653 | 0.847 | 0.826 | 0.853 | 0.877 | 0.878 | 0.760 |
| | | | | *Parameter-Efficient Fine Tuning (LoRA)* | | | | | | |
| | Dense Model | 0.607 | 0.565 | 0.681 | 0.853 | 0.963 | 0.846 | 0.909 | 0.896 | 0.790 |
| 2 | GradPruner | 0.588 | 0.565 | 0.659 | 0.840 | 0.963 | 0.832 | 0.893 | 0.896 | 0.780 |
| | w/o Merging | 0.561 | 0.540 | 0.632 | 0.827 | 0.911 | 0.799 | 0.858 | 0.860 | 0.749 |
| 1 | GradPruner | 0.606 | 0.566 | 0.677 | 0.849 | 0.960 | 0.846 | 0.905 | 0.902 | 0.789 |
| | w/o Merging | 0.590 | 0.544 | 0.657 | 0.826 | 0.933 | 0.819 | 0.875 | 0.877 | 0.765 |

Table 4: Experimental results on the impact of the number of merged layers on downstream task accuracy (LoRA fine-tuning).

3) BillSum (Kornilova & Eidelman, 2019) is the first dataset for summarization of US Congressional and California state bills, which is in the financial domain.

4) fingpt-sentiment-train Yang et al., 2023 is a financial sentiment analysis question-answering dataset containing 76,000 samples. We randomly selected 40,000 samples as the training set.

5) HellaSwag (Zellers et al., 2019) is a challenging dataset for evaluating commonsense NLI that is especially hard for state-of-the-art models, though its questions are trivial for humans (¿95% accuracy).

6) WinoGrande (Sakaguchi et al., 2021)consists of 10.2k training samples for cloze tests, measuring performance with accuracy.

7) ARC (ARC-Easy) (Clark et al., 2018) is a multiple-choice question-answering benchmark designed to test the model's ability to reason about scientific knowledge.

8) PIQA (Bisk et al., 2020) is a physical commonsense reasoning dataset that is designed to test the model's ability to build, craft, or manipulate objects using everyday physical knowledge.

## C.2 Training Details

There are two ways of fine-tuning. One is Full Fine-Tuning (FFT). The other is Parameter-Efficient Fine-Tuning, and we adopt the LoRA method. During training, we set the learning rate to 1e-5 and the batch size to 64. Each dataset was trained for 3 epochs. The AdamW optimizer was used for fine-tuning. We employed SWIFT as the training platform and vLLM for inference. We set the sparsity ratio during merging to 80% and pruned 13 layers (approximately 40% of the total parameters), three of which applied the layer merging technique.

## D Supplementary Ablation Study

**Perform Only the Layer Pruning in GradPruner:** Figure 8 presents the results of using only the layer pruning strategy. For Mistral-7B, pruning up to 11 layers has minimal impact on downstream task accuracy. However, beyond 11 layers, accuracy begins to decline significantly, as increasingly critical layers are removed.

**The Impact of the Number of Merged Layers on Accuracy:** Table 4 demonstrates the impact on downstream task accuracy after incorporating the layer merging algorithm on top of layer pruning.

From the table, we can observe that for both models, whether fine-tuned using FFT or LoRA, layer merging significantly improves accuracy. For Mistral-7B, merging 1 layer after pruning 11 layers aligns its accuracy with the dense model. Although merging 2 layers introduces some accuracy differences, the model still maintains relatively high performance.

**Replacing Layer Pruning with Kernel Pruning:** Readers may wonder why GradPruner focuses on pruning layers rather than adopting a finer-grained approach, such as pruning kernels (rows or columns) within parameter matrices. To address this question, we applied pruning to the Hidden Size (HZ) within the GradPruner framework. Specifically, we used IGIA-Matrix to compute the importance of each hidden size, pruned the less important parts, and merged the pruned hidden sizes with the unpruned ones. The pruning rate was set to 40%. The experimental results are shown in Figure 9 below. Compared to layer pruning, using the finer-grained kernel pruning leads to a noticeable decline in accuracy. We believe this phenomenon is mainly caused by two reasons: 1) The importance of layers may take precedence over that of kernels, and pruning kernels in critical layers could lead to a drop in accuracy. 2) Pruning hidden sizes requires merging a larger number of hidden sizes, which is less conducive to maintaining accuracy.

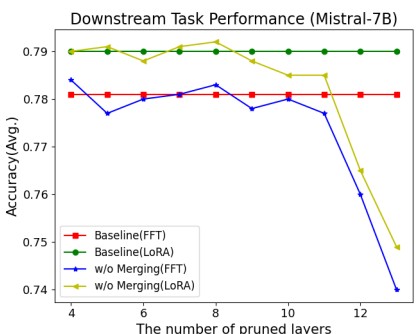

Figure 8: Experimental results of performing Only the layer pruning in GradPruner.

Figure 9: Experimental results of replacing layer pruning with kernel pruning in Grad-Pruner. We report the average accuracy across eight datasets.

**Layer-importance estimation for tasks with varying gradient noise:** To further explore the different behaviors of layer importance estimation for downstream tasks under various noise levels, we supplemented the following experiments: Based on the WinoGrande dataset (original size 10.2k), we randomly sampled 1%, 10%, 30%, and 100% of its data as training sets, systematically evaluating the different behaviors of layer importance estimation under different training data sizes (different gradient variances), paying particular attention to whether early gradient signals remain reliable with very small amounts of data. The relevant results are shown in Tab. 5.

Table 5: Ablation experiments with different training data sizes were conducted on the WinoGrande dataset, based on Llama 3.1-8B (Full Fine-tuning). The Dense Model represents the accuracy of the unpruned model after fine-tuning on downstream tasks.

| Training data ratio and size | 1% (102) | 10% (1.02k) | 30% (3.06k) | 100% (10.2k) |
|---|---|---|---|---|
| Dense Model | 0.853 | 0.863 | 0.866 | 0.868 |
| GradPruner (ours) | 0.842 | 0.857 | 0.862 | 0.861 |

Experiments show that even with a significant reduction in training data, our layer importance estimation maintains good stability. Only when the data volume is extremely low (e.g., only about 100 samples) does the model performance fluctuate slightly, but the overall trend remains robust. This indicates that our method's reliance on early-step gradients is reasonable under typical data scales.

**Performance across Sparsity Ratio:** To comprehensively evaluate the relative advantages of Grad-Pruner across different sparsity levels, Tab. 6 extends the main experiment at 40% sparsity by additional comparisons against baseline methods at 30% and 20% sparsity

We did not consider sparsity levels higher than 40% because, under which settings most methods—including GradPruner—exhibit significant performance degradation. Existing baseline methods already struggle to maintain reasonable accuracy at 40% sparsity, and increasing sparsity further would exacerbate this performance drop, rendering comparisons at higher sparsity levels uninformative.

Table 6: Comparative experiments between our method and other approaches under different sparsity levels, using the Llama3.1-8B model with full fine-tuning on the HellaSwag dataset. "Dense Model" denotes the accuracy of the unpruned model after fine-tuning on the downstream task.

| Methods | Dense Model | LLMPruner | Laco | MINITRON | SAT | GradPruner (ours) |
|---|---|---|---|---|---|---|
| Sparsity Ratio (30%) | 0.943 | 0.916 | 0.923 | 0.923 | 0.934 | 0.940 |
| Sparsity ratio(20%) | 0.943 | 0.927 | 0.936 | 0.929 | 0.940 | 0.942 |

The experimental results in Tab. 6 show the following:

At 30% sparsity, GradPruner continues to maintain a clear advantage over all baseline methods. At 20% sparsity (i.e., under milder pruning), some methods—such as Laco and SAT—also demonstrate strong performance; however, our method remains competitive and exhibits consistently robust overall performance. These findings indicate that the advantages of GradPruner are not limited to moderate sparsity levels (e.g., 40%) and can be generalized to lighter pruning settings, demonstrating its effectiveness and robustness across a range of sparsity regimes.

**Performance across Model Scales:**To verify the applicability of GradPruner across different model scales, Tab. 7 shows experiments on both smaller (approximately 1B) and larger (approximately 13B) models, using Llama-3.2-1B and Llama-2-13B as testbeds, respectively. The experiments demonstrate that our method performs effectively across both model scales: it not only maintains strong performance on the larger Llama-2-13B but also exhibits even more pronounced advantages on the smaller Llama-3.2-1B. This indicates that GradPruner's pruning mechanism is highly scalable, with its effectiveness consistently extending from 1B-scale models to those exceeding 10B parameters, showcasing strong generality.

Table 7: We evaluate GradPruner on models of different sizes: for Llama-3.2-1B (originally 16 layers), we prune 6 layers; for Llama-2-13B (originally 40 layers), we prune 16 layers. "Dense Model" denotes the accuracy of the unpruned model after fine-tuning on the downstream task.

| Datasets | MedMCQA | FinGPT | HellaSwag |
|---|---|---|---|
| Llama-3.2-1B(Dense Model) | 0.434 | 0.543 | 0.584 |
| Llama-3.2-1B(GradPruner) | 0.434 | 0.538 | 0.585 |
| Llama-2-13B(Dense Model) | 0.542 | 0.806 | 0.907 |
| Llama-2-13B(GradPruner) | 0.532 | 0.798 | 0.894 |

**The Impact of the Number of Initial Fine-tuning Steps:** We conducted ablation experiments on the HellaSwag dataset with different gradient accumulation steps. The steps we selected were 0.02%, 0.05%, 0.08%, and 1%. The experimental results are shown in Tab. 8. Experimental results show that our method has certain requirements for the number of gradient accumulation steps; performance will decrease when the number is less than 0.05%, but this requirement is not stringent—experiments show that only about 1% of the training steps are needed to achieve good alignment results and stable performance.

Table 8: Ablation experiments with different gradient accumulation steps were conducted on the HellaSwag dataset, based on Llama 3.1-8B (Full Fine-tuning). The Dense Model represents the accuracy of the unpruned model after fine-tuning on downstream tasks.

| | 0.02% | 0.05% | 0.08% | 1% |
|---|---|---|---|---|
| Dense Model | 0.943 | 0.943 | 0.943 | 0.943 |
| GradPruner (ours) | 0.911 | 0.929 | 0.939 | 0.939 |

To check the effectiveness with even fewer gradient accumulation steps, we conducted ablation experiments on the HellaSwag dataset with different gradient accumulation steps. The steps we selected were 0.02%, 0.05%, 0.08%, and 1%. The experimental results are shown in Tab. 9.

Experimental results show that our method has certain requirements for the number of gradient accumulation steps; performance will decrease when the number is less than 0.05%, but this requirement is not stringent—experiments show that only about 1% of the training steps are needed to achieve good alignment results and stable performance.

Table 9: Ablation experiments with different gradient accumulation steps were conducted on the HellaSwag dataset using the Llama 3.1-8B model. The Dense Model represents the accuracy of the unpruned model after fine-tuning on downstream tasks.

|  | 0.02% | 0.05% | 0.08% | 1% |
|---|---|---|---|---|
| Dense Model | 0.943 | 0.943 | 0.943 | 0.943 |
| GradPruner (ours) | 0.911 | 0.929 | 0.939 | 0.939 |

## D.1 STABILITY OF EARLY-STEP GRADIENTS

To further explore the stability of early-step gradients, we added the following experiments: Based on the WinoGrande dataset (original size 10.2k), we randomly sampled 1%, 10%, 30%, and 100% of its data as training sets, systematically evaluating the stability of early-step gradients under different training data sizes, paying particular attention to whether the early gradient signals remain reliable with extremely small datasets. The relevant results are shown in the Tab. 10.

Experiments show that even with a significant reduction in training data, layer importance estimation based on early-step gradients maintains good stability. Only when the amount of data is extremely low (e.g., only about 100 samples) does the model performance fluctuate slightly, but the overall trend remains robust. This indicates that our method's reliance on early-step gradients is reasonable under typical data scales.

Table 10: Ablation experiments with different training data sizes were conducted on the WinoGrande dataset, based on Llama 3.1-8B (Full Fine-tuning). The Dense Model represents the accuracy of the unpruned model after fine-tuning on downstream tasks.

| training data ration and size | 1% (102) | 10% (1.02k) | 30% (3.06k) | 100% (10.2k) |
|---|---|---|---|---|
| Dense Model | 0.853 | 0.863 | 0.866 | 0.868 |
| GradPruner (ours) | 0.842 | 0.857 | 0.862 | 0.861 |

## D.2 EFFICIENCY FROM A FLOPS PERSPECTIVE

Tab. 11 above shows a comparison of the efficiency of the original llama3.1-8B model and the model pruned using GradPruner. During testing, we used an input length of 128 tokens. The results show that the pruned model reduced the number of parameters by approximately 38%, and saved approximately 40% in FLOPs and MACs.

## D.3 GENERALIZATION IN CROSS-DATASET SCENARIOS

To verify the generalization ability of the method in cross-dataset scenarios, we supplemented it with cross-dataset evaluation experiments within the domain: pruning and fine-tuning were per-

Table 11: Two metrics, FLOPs and MACs, represent the efficiency of GradPruner

| llama3.1-8B | FLOPs (TFLOPS) | MACs (GMACs) | Params (B) |
|---|---|---|---|
| original | 1.79 | 893.35 | 7.5 |
| pruned | 1.06 | 530.43 | 4.67 |

formed using the original medical domain training set MedMCQA from the paper, and testing was conducted on the medical subset clinical knowledge in MMLU. The results are shown in Tab. 12. Despite a slight performance drop compared to Dense models in cross-dataset settings (i.e., training and testing distributions are not perfectly consistent), GradPruner still maintains a significant accuracy advantage over other existing pruning methods.

Table 12: Cross-dataset evaluations in MedMCQA, based on Llama 3.1-8B (Full Fine-tuning). Training data is MedMCQA, and test data is clinical knowledge. The Dense Model represents the accuracy of the unpruned model after fine-tuning on downstream tasks.

| Methods | Dense Model | LLMPruner | Laco | MINITRON | SAT | GradPruner (ours) |
|---------|-------------|-----------|------|----------|-----|-------------------|
|         | 0.735       | 0.688     | 0.711 | 0.703   | 0.709 | 0.717           |

### D.4 GENERALIZATION WITH OTHER ADAPTER METHODS

To verify the stability of the IGIA matrix under different adapter methods, we conducted additional experiments: we replaced the strategy in GradPruner that originally used LoRA fine-tuning to evaluate the importance of pre-trained parameters with QLoRA and DoRA, respectively. The relevant results are summarized in Tab. 13. As the experimental results show, the performance of DoRA is comparable to LoRA, with almost no significant difference; however, a slight decrease in accuracy occurs when using QLoRA. This phenomenon can be attributed to the technical characteristics of the two methods: DoRA decomposes the pre-trained weights, achieving an optimization behavior closer to full parameter fine-tuning than LoRA, thus better evaluating the importance of pre-trained parameters; in contrast, QLoRA introduces quantization operations on the pre-trained weights on top of LoRA, which introduces quantization noise, leading to some bias in the estimated parameter importance.

Table 13: Stability tests were conducted on the HellaSwag dataset using different adapters based on Llama 3.1-8B (Full Fine-tuning). The Dense Model represents the accuracy of the unpruned model after fine-tuning on downstream tasks.

|                  | LoRA  | DoRA  | QLoRA |
|------------------|-------|-------|-------|
| Dense Model      | 0.943 | 0.943 | 0.943 |
| GradPruner (ours)| 0.939 | 0.941 | 0.925 |

