# OpenReview forum: "GradPruner: Gradient-guided Layer Pruning Enabling Efficient Fine-Tuning and Inference for LLMs"
_ICLR.cc/2026/Conference — ICLR 2026 Poster_

### Official Review · Reviewer_QPt3 · 2025-10-31

**Soundness:** 3
**Presentation:** 2
**Contribution:** 3
**Rating:** 6
**Confidence:** 3

**Summary:**

GradPruner uses the IGIA-Matrix computed from the early stages of fine-tuning (within the first 1% of steps) to evaluate the importance of parameters and layers. Layers with low importance scores are pruned, and to further retain performance under high pruning rates, the parameters of pruned layers are merged into the retained layer using a sign-based merging rule. Experiments on two LLMs across eight datasets show that GradPruner achieves approximately 40% parameter reduction with less than 1% average accuracy drop, while also significantly reducing fine-tuning time, inference latency, and memory consumption compared to representative pruning baselines.

**Strengths:**

* GradPruner jointly improves both training and inference efficiency. Unlike many pruning approaches that focus only on inference speed-ups, GradPruner is explicitly designed to reduce fine-tuning time and memory consumption as well. I believe this aligns well with the practical need for LLMs to quickly adapt to new downstream tasks.
* GradPruner is simple to understand yet conceptually novel. The paper introduces a new gradient-based importance metric computed from less than 1% of the early fine-tuning steps, which effectively reduces the computational cost of pruning. In addition, it seems like the proposed pruning strategy is generally applicable to all transformer-based architectures.
* The experiments are comprehensive. Evaluations are conducted on two representative LLMs across eight datasets, which enhances the reliability of the results. The paper also provides detailed ablation studies that clearly demonstrate the contribution of each component.

**Weaknesses:**

* Limited theoretical grounding. Lines 201–208 simulate the gradient of W via a matrix multiplication, yet the rationale for this simulation is not theoretically justified. In addition, the sign-based merging rule in Equation (5) is presented as a heuristic with little theoretical explanation. Clearer derivations would strengthen the contribution.
* Concern over the stability of early-step gradients. The method relies on gradient statistics collected within the first 1% of fine-tuning steps to estimate layer importance, yet such early gradients may not always provide a stable or reliable signal. For example, in small or noisy datasets, gradient variance can be high, potentially causing the IGIA-Matrix to mis-rank layer importance and resulting in suboptimal pruning decisions.
* Incomplete efficiency metrics. The paper aims to make layer pruning more efficient and reports average time and parameter reduction. Including FLOPs would provide a more standardized and hardware-agnostic measure of computational efficiency.
* Clarity of the method description could be improved. The paper does not explicitly detail how each transformer sub-module is handled during merging. It appears that each linear sub-module of a pruned layer is merged into the corresponding sub-module of the preceding retained layer, but this should be stated unambiguously. Moreover, Equation (5), which specifies merging based on sign agreement, is difficult to parse. A more thorough explanation and a small illustrative example would aid the reader's understanding.

**Questions:**

* Can you clarify the exact computation and shape alignment for simulating the gradient of W via LoRA gradients?
* Why is raw summation chosen for IGIA-matrix aggregation across all linear layers, rather than normalization (e.g., mean or norm per parameter or per layer size)?
* Could the authors elaborate on the layer merging procedure, perhaps with an example?
* Can the approach still be effective with even fewer gradient accumulation steps (e.g., less than 0.02% steps)？

---

> ### Author Response · Authors · 2025-11-17
> **Response to Reviewer QPt3 [1/2]**
>
> We sincerely thank you for taking the time to review our paper and for providing valuable comments and suggestions.
>
> ## Question: Lines 201–208 simulate the gradient of W via a matrix multiplication, yet the rationale for this simulation is not theoretically justified.
>
> In full-parameter fine-tuning, the original weight matrix $W\in\mathbb{R}^{d×d}$ directly participates in forward and backward propagation, and the model's behavior is equivalent to $\widetilde{W}=W+W\^\*$, where $W\^\*$ is the parameter increment during fine-tuning, and its gradient is $\nabla_{W} L$. However, in LoRA, we freeze the original weights W and introduce a low-rank trainable increment: $\triangle W =BA$, where $A\in\mathbb{R}^{r×d}, B\in\mathbb{R}^{d×r}$. At this point, forward propagation becomes $h = (W + BA)x$. In LoRA fine-tuning, the model's behavior is equivalent to using a perturbed weight $\widetilde{W}=W+(BA)\^\*$, where $(BA)\^\*$ represents the increment learned by LoRA. Therefore, from the perspective of first-order Taylor expansion, the increment $(BA)\^\*$ learned by LoRA will approximate the optimal update direction $\delta W^{\ast}\approx -\eta \nabla_{W} L$ under full fine-tuning as closely as possible. This point was also proposed in the LoRA[1] paper (The study presented in LoRA [1] suggests that LoRA can be considered a general approximation of full fine-tuning.). Based on the above analysis, we propose the following approximation $\nabla_{W} L \overset{\text{sim}}{=} \nabla_{W_{B}} \cdot \nabla_{W_{A}} L$ (Formula 2 in the paper).
>
>
>
> [1] LORA: LOW-RANK ADAPTATION OF LARGE LANGUAGE MODELS
>
>
> ## Question: In addition, the sign-based merging rule in Equation (5) is presented as a heuristic with little theoretical explanation.
>
> In our response to all reviewers, we explained "Theory of Merging in GradPruner". For specific theoretical details, please refer to the response at the top. According to our theoretical explanation,
>
> GradPruner’s importance criterion simultaneously leverages (1) magnitude—quantified by the squared gradient—and (2) directional consistency—captured by the agreement in sign between the two weights—as complementary indicators of parameter significance.
>
> The perspectives of using gradient magnitude for assessing parameter importance refer to previous work [1][4], and the idea of ​​using sign consistency refers to previous work [2][3]. GradPruner unifies both perspectives into a single adaptive framework.
>
> [1] Merging layers with Fisher-Weighted Averaging
>
> [2] NegMerge: Sign-Consensual Weight Merging for Machine Unlearning
>
> [3] TIES-Merging: Resolving Interference When Merging Models
>
> [4] Model Merging by Uncertainty-Based Gradient Matching
>
>
>
> ## Question: Concern over the stability of early-step gradients. The method relies on gradient statistics collected within the first 1\% of fine-tuning steps to estimate layer importance, yet such early gradients may not always provide a stable or reliable signal.
>
> To further explore the stability of early-step gradients, we added the following experiments: Based on the WinoGrande dataset (original size 10.2k), we randomly sampled 1\%, 10\%, 30\%, and 100\% of its data as training sets, systematically evaluating the stability of early-step gradients under different training data sizes, paying particular attention to whether the early gradient signals remain reliable with extremely small datasets. The relevant results are shown in the table below.
>
> Table: Ablation experiments with different training data sizes were conducted on the WinoGrande dataset, based on Llama 3.1-8B (Full Fine-tuning). The Dense Model represents the accuracy of the unpruned model after fine-tuning on downstream tasks.
>
> | Training data ratio and size | 1% (102) | 10% (1.02k) | 30% (3.06k) | 100% (10.2k) |
> |--|---|--|-|-|
> | Dense Model  | 0.853    | 0.863   | 0.866 | 0.868  |
> | GradPruner (ours)   | 0.842    | 0.857   | 0.862   | 0.861        |
>
> Experiments show that even with a significant reduction in training data, layer importance estimation based on early-step gradients maintains good stability. Only when the amount of data is extremely low (e.g., only about 100 samples) does the model performance fluctuate slightly, but the overall trend remains robust. This indicates that our method's reliance on early-step gradients is reasonable under typical data scales.
>
>
> ## Question: Incomplete efficiency metrics. The paper aims to make layer pruning more efficient and reports average time and parameter reduction.
>
> | llama3.1-8B | FLOPs (TFLOPS) | MACs (GMACs) | Params (B) |
> |-|--|-|-|
> | original    | 1.79 | 893.35  | 7.5  |
> | pruned      | 1.06 | 530.43  | 4.67 |
>
> The table above shows a comparison of the efficiency of the original llama3.1-8B model and the model pruned using GradPruner. During testing, we used an input length of 128 tokens. The results show that the pruned model reduced the number of parameters by approximately 38\%, and saved approximately 40\% in FLOPs and MACs.

---

> > ### Author Response · Authors · 2025-11-17
> > **Response to Reviewer QPt3 [2/2]**
> >
> > ## Question: Clarity of the method description could be improved. The paper does not explicitly detail how each transformer sub-module is handled during merging. Moreover, Equation (5), which specifies merging based on sign agreement, is difficult to parse. A more thorough explanation and a small illustrative example would aid the reader's understanding.
> >
> > We provide a concrete example in Figure 4 of the paper: where $W_j$ represents the retained layer, and $W_{j+1}, W_{j+2}$ represent the layers to be pruned. The entire merging process consists of three steps:
> >
> > (1) First, $W_{j+1}, W_{j+2}$ are sparsified. The red parts in the figure represent weights that will be set to zero.
> >
> > (2) Subsequently, based on sign consistency (as described in Equation (5)), the mergeable positions are further screened. The positions marked with a cross in the figure represent weights that cannot be merged due to sign inconsistency.
> >
> > (3) Finally, the remaining non-zero weights in $W_{j+1}, W_{j+2}$ are merged into the submodule corresponding to the previous retained layer $W_j$.
> >
> > To improve readability, we will more clearly indicate the above three steps in Figure 4, so that readers can intuitively understand our layer merging mechanism through this concise example. Furthermore, as the reviewers pointed out, lines 301–303 of the paper clearly state: “The linear submodules in each pruned layer will be merged into the submodules corresponding to the previous retained layer.” We will further strengthen this description in the main text to eliminate any possible ambiguity.
> >
> > ## Question: Can you clarify the exact computation and shape alignment for simulating the gradient of W via LoRA gradients?
> >
> > We consider a linear layer with a weight matrix of $W\in\mathbb{R}^{m×n}$ and a corresponding loss gradient of $\nabla_{W} L \in\mathbb{R}^{m×n}$. In LoRA fine-tuning, we introduce two trainable low-rank matrices $B\in\mathbb{R}^{m×r}$ and $A\in\mathbb{R}^{r×n}$, and freeze the original weights $W$ during training. The forward propagation calculation is: $h = (W + BA)x$. During the backpropagation, we obtain the gradients of $\nabla_{W_A} L \in\mathbb{R}^{r×n}$ and $\nabla_{W_A} L \in\mathbb{R}^{m×r}$, respectively. To simulate the gradient of the original weight $W$, we compute the matrix product of these two gradients: $\nabla_{W_B} L \cdot \nabla_{W_A} L \in\mathbb{R}^{m×n}$, which has the same shape as $\nabla_{W} L$.
> >
> >
> > ## Question: Why is raw summation chosen for IGIA-matrix aggregation across all linear layers, rather than normalization (e.g., mean or norm per parameter or per layer size)?
> >
> > Because the IGIA-matrix contains the parameter importance of all matrices in the model, we evaluate the relative importance of the current layer to other layers by adding the importance of all parameter matrices in a certain layer. If averaging and normalizing all parameters in this layer does not affect the ranking of the relative importance scores between the current layer and the other layers, we choose to perform the simpler operation of direct addition.
> >
> > ## Question: Could the authors elaborate on the layer merging procedure, perhaps with an example?
> >
> > Regarding the specific process of layer merging, we have already explained it in our response to point 4 of Weaknesses, and a concise example is provided in Figure 4 of the paper to aid understanding.
> >
> > ## Question: Can the approach still be effective with even fewer gradient accumulation steps (e.g., less than 0.02\% steps)
> >
> > To address your question, we conducted ablation experiments on the HellaSwag dataset with different gradient accumulation steps. The steps we selected were 0.02\%, 0.05\%, 0.08\%, and 1\%. The experimental results are shown in the table below.
> >
> > Table: Ablation experiments with different gradient accumulation steps were conducted on the HellaSwag dataset using the Llama 3.1-8B model. The Dense Model represents the accuracy of the unpruned model after fine-tuning on downstream tasks.
> >
> > |             | 0.02% | 0.05% | 0.08% | 1%   |
> > |-------------|-------|-------|-------|------|
> > | Dense Model | 0.943 | 0.943 | 0.943 | 0.943 |
> > | GradPruner (ours) | 0.911 | 0.929 | 0.939 | 0.939 |
> >
> >
> > Experimental results show that our method has certain requirements for the number of gradient accumulation steps; performance will decrease when the number is less than 0.05\%, but this requirement is not stringent—experiments show that only about 1\% of the training steps are needed to achieve good alignment results and stable performance.

---

> > ### Comment · Reviewer_QPt3 · 2025-11-24
> >
> > Thank you for the author's response. My score is already positive, and I will keep it.

---

### Official Review · Reviewer_87rX · 2025-11-01

**Soundness:** 3
**Presentation:** 3
**Contribution:** 4
**Rating:** 6
**Confidence:** 4

**Summary:**

This paper proposes GradPruner, a gradient-guided structured pruning framework designed to enhance both fine-tuning and inference efficiency for large language models (LLMs). Unlike most existing structured pruning methods that rely on calibration data or additional distillation steps, GradPruner leverages initial gradient information obtained during the first few LoRA fine-tuning steps to estimate parameter and layer importance.

The key innovation lies in computing an Initial Gradient Information Accumulation Matrix (IGIA-Matrix) to quantify layer importance early in training. Based on this metric, GradPruner performs layer-level pruning followed by a layer merging step that sparsifies and merges pruned layers into adjacent retained layers while resolving sign conflicts to reduce destructive interference.

Empirically, GradPruner achieves a 40% reduction in parameters with only 0.99% loss in downstream accuracy, tested on two LLMs (LLaMA3.1-8B and Mistral-7B) across eight diverse benchmarks spanning medical, financial, and general reasoning tasks. It shows consistent performance improvements over strong structured pruning baselines (LLM-Pruner, LaCo, SAT, APT, and MINITRON), and reduces both training and inference time/memory by over 35%.

**Strengths:**

1. The use of IGIA-Matrix computed from <1% of training steps is original and empirically justified by gradient sensitivity analysis.


2. The proposed sign-consistent merging technique effectively preserves accuracy even under 40% pruning.


3. The authors test across multiple domains, model sizes, and fine-tuning regimes with strong baselines, demonstrating robustness.


4. Substantial reductions in both training and inference costs (~35–40%) while maintaining accuracy are practically valuable.


5. The paper analyzes pruning ratios, merging counts, and alternatives such as kernel pruning and weighted averaging.

**Weaknesses:**

1. While the empirical gradient correlation study is convincing, the paper lacks a deeper theoretical analysis of why early gradient accumulation correlates with long-term importance, beyond empirical observation.


2. The method assumes access to LoRA gradients and may not generalize to non-LoRA or adapter-free fine-tuning setups.


3. The layer-importance estimation could behave differently for tasks with varying gradient noise; this is not fully explored.

**Questions:**

1. How sensitive is GradPruner’s performance to the number of initial fine-tuning steps ttt?


2. Have the authors tested GradPruner when using other adapter methods (e.g., QLoRA, DoRA) to verify whether IGIA-Matrix remains stable?


3. In the merging phase, how is the sparsity rate ppp selected? Could adaptive sparsity (learned from IGIA statistics) yield further improvements?


4. Does GradPruner preserve the same inference graph (number of layers) after merging, or does merging affect sequence length/runtime at deployment?


5. Can GradPruner be integrated with post-training quantization or low-rank compression? If so, how does the gradient-based importance interact with quantization noise?

---

> ### Author Response · Authors · 2025-11-17
> **Response to Reviewer 87rX [1/2]**
>
> We sincerely thank you for taking the time to review our paper and for providing valuable comments and suggestions.
>
>
> ## Question: While the empirical gradient correlation study is convincing, the paper lacks a deeper theoretical analysis of why early gradient accumulation correlates with long-term importance, beyond empirical observation.
>
>  During the initial phase of fine-tuning, the model undergoes rapid adaptation to the downstream task (as shown in Figure 1), and the dominant accumulated gradient directions reflect the most informative features for that task. This aligns with findings in the Lottery Ticket Hypothesis (LTH) [1],  which indicates that "A randomly-initialized, dense neural network contains a subnetwork that is initialized such that—when trained in isolation—it can match the test accuracy of the original network after training for at most the same number of iterations". To find such subnetworks, **LTH trains a randomly initialized network for a few steps and then prunes it according to the magnitude of the weights.  Theoretically, our pruning according to IGIA-Matrix  (as shown in eq.3, calculated by accumulating gradients in early steps)  is theoretically equal to the pruning method in LTH.** Therefore, theoretically, our pruning method is an effective way to identify important weights.
>
> [1] The Lottery Ticket Hypothesis: Finding Sparse, Trainable Neural Networks
>
>
>
> ## Question: The method assumes access to LoRA gradients and may not generalize to non-LoRA or adapter-free fine-tuning setups.
>
> The core of our method is: We obtain the importance scores of each parameter through LoRA fine-tuning on the large model before pruning, and then perform layer pruning accordingly. Therefore, this method only requires that the original large model can support LoRA fine-tuning to calculate importance, and has no restrictions on the fine-tuning method used in the pruned model (whether it's LoRA, other adapter methods, or full-parameter fine-tuning). In other words, the pruning stage relies on LoRA gradients, but the subsequent fine-tuning stage can flexibly adapt to various training paradigms, exhibiting good compatibility.
>
> ## Question: The layer-importance estimation could behave differently for tasks with varying gradient noise; this is not fully explored.
>
> To further explore the different behaviors of layer importance estimation for downstream tasks under various noise levels, we supplemented the following experiments: Based on the WinoGrande dataset (original size 10.2k), we randomly sampled 1\%, 10\%, 30\%, and 100\% of its data as training sets, systematically evaluating the different behaviors of layer importance estimation under different training data sizes (different gradient variances), paying particular attention to whether early gradient signals remain reliable with very small amounts of data. The relevant results are shown in the table below.
>
> Table: Ablation experiments with different training data sizes were conducted on the WinoGrande dataset, based on Llama 3.1-8B (Full Fine-tuning). The Dense Model represents the accuracy of the unpruned model after fine-tuning on downstream tasks.
>
> | Training data ratio and size | 1% (102) | 10% (1.02k) | 30% (3.06k) | 100% (10.2k) |
> |---------|----------|-------------|-------------|--------------|
> | Dense Model                 | 0.853    | 0.863       | 0.866       | 0.868        |
> | GradPruner (ours)           | 0.842    | 0.857       | 0.862       | 0.861        |
>
> Experiments show that even with a significant reduction in training data, our layer importance estimation maintains good stability. Only when the data volume is extremely low (e.g., only about 100 samples) does the model performance fluctuate slightly, but the overall trend remains robust. This indicates that our method's reliance on early-step gradients is reasonable under typical data scales.
>
> ## Question: How sensitive is GradPruner’s performance to the number of initial fine-tuning steps?
>
> To address your question, we conducted ablation experiments on the HellaSwag dataset with different gradient accumulation steps. The steps we selected were 0.02\%, 0.05\%, 0.08\%, and 1\%. The experimental results are shown in the table below.
>
> Table: Ablation experiments with different gradient accumulation steps were conducted on the HellaSwag dataset, based on Llama 3.1-8B (Full Fine-tuning). The Dense Model represents the accuracy of the unpruned model after fine-tuning on downstream tasks.
>
> |   | 0.02% | 0.05% | 0.08% | 1%   |
> |----|---|----|----|----|
> | Dense Model | 0.943 | 0.943 | 0.943 | 0.943 |
> | GradPruner (ours) | 0.911 | 0.929 | 0.939 | 0.939 |
>
> Experimental results show that our method has certain requirements for the number of gradient accumulation steps; performance will decrease when the number is less than 0.05\%, but this requirement is not stringent—experiments show that only about 1\% of the training steps are needed to achieve good alignment results and stable performance.

---

> > ### Author Response · Authors · 2025-11-17
> > **Response to Reviewer 87rX [2/2]**
> >
> > ## Question: Have the authors tested GradPruner when using other adapter methods  to verify whether IGIA-Matrix remains stable?
> >
> > To verify the stability of the IGIA matrix under different adapter methods, we conducted additional experiments: we replaced the strategy in GradPruner that originally used LoRA fine-tuning to evaluate the importance of pre-trained parameters with QLoRA and DoRA, respectively. The relevant results are summarized in the table below.
> >
> > As the experimental results show, the performance of DoRA is comparable to LoRA, with almost no significant difference; however, a slight decrease in accuracy occurs when using QLoRA. This phenomenon can be attributed to the technical characteristics of the two methods: DoRA decomposes the pre-trained weights, achieving an optimization behavior closer to full parameter fine-tuning than LoRA, thus better evaluating the importance of pre-trained parameters; in contrast, QLoRA introduces quantization operations on the pre-trained weights on top of LoRA, which introduces quantization noise, leading to some bias in the estimated parameter importance.
> >
> > Table: Stability tests were conducted on the HellaSwag dataset using different adapters based on Llama 3.1-8B (Full Fine-tuning). The Dense Model represents the accuracy of the unpruned model after fine-tuning on downstream tasks.
> >
> > |                   | LoRA  | DoRA  | QLoRA |
> > |-------------------|-------|-------|-------|
> > | Dense Model       | 0.943 | 0.943 | 0.943 |
> > | GradPruner (ours) | 0.939 | 0.941 | 0.925 |
> >
> >
> > ## Question: In the merging phase, how is the sparsity rate selected? Could adaptive sparsity (learned from IGIA statistics) yield further improvements?
> >
> > Regarding the choice of sparsity rate, we conducted systematic ablation experiments in Section 6 and Figure 6 of the paper. Analysis of the experimental results shows that both excessively high and low sparsity rates negatively impact task accuracy to varying degrees. Considering both model compression and performance preservation, we ultimately selected 80\% as the default sparsity rate.
> >
> > Furthermore, we fully agree with the reviewers' observations: adopting an adaptive sparsity strategy is expected to better balance two key objectives: (1) minimizing interference with retained layer parameters during layer merging; and (2) more accurately preserving highly important parameters in pruned layers. This mechanism indeed has the potential to further improve model accuracy. We are very grateful for this insightful suggestion and will explore the design and optimization of adaptive sparsity rates in future work.
> >
> > ## Question4: Does GradPruner preserve the same inference graph (number of layers) after merging, or does merging affect sequence length/runtime at deployment?
> >
> > Our proposed layer merging method aims to further enhance the potential of model pruning. Experiments show that for the Llama3.1-8B model (32 layers), without layer merging, a maximum of 10 layers can be pruned to ensure downstream task accuracy, resulting in an inference speedup of approximately 30\%. However, by introducing our merging strategy, an additional 3 layers can be pruned. Specifically, these 3 layers are completely removed from the inference computation graph (i.e., no longer participating in forward propagation), directly reducing model depth and inference latency, resulting in an inference speedup of approximately 40\%. Simultaneously, their parameter information is fused into the retained layers through the proposed merging mechanism to preserve the expressive power of the original model as much as possible. Therefore, our merging technique is essentially designed to support a higher degree of structured pruning, and it does indeed reduce the number of network layers and deliver a real inference speedup during deployment.
> >
> > ## Question: Can GradPruner be integrated with post-training quantization or low-rank compression? If so, how does the gradient-based importance interact with quantization noise?
> >
> > Our method's first stage involves fine-tuning the large model before pruning using LoRA to obtain importance scores for each parameter, and then performing layer pruning accordingly. The second stage involves post-training the pruned model on downstream tasks. In principle, our method primarily improves the training phase. After training a model using our method, post-training quantization can be applied to this model without any workflow conflict. However, we are unsure whether post-training quantization on a model trained using our method will significantly alter the performance of the quantized model. This is an interesting direction that we will explore further in future work.

---

### Official Review · Reviewer_RvRE · 2025-11-01

**Soundness:** 2
**Presentation:** 1
**Contribution:** 3
**Rating:** 2
**Confidence:** 4

**Summary:**

The paper proposes **GradPruner**, a lora-based gradient-guided **layer pruning + sign-based layer merging** method for efficient fine-tuning and inference. The method (i) computes an **Initial Gradient Information Accumulation (IGIA)** matrix from early-step LoRA gradients to score parameter/layer importance, (ii) prunes layers with low summed IGIA (over linear sublayers), and (iii) **sparsify-then-merge** pruned layers into the preceding retained layer using a **sign-consistency rule**. Experiments on two LLMs (Llama-3.1-8B, Mistral-7B) and eight datasets report ~**40% sparsity** with ~**0.99%** average degradation; the headline comparison (Table 1) is at **40% sparsity**.

**Strengths:**

- **Simple, pragmatic pipeline.** Early-step gradient accumulation → IGIA-based layer scoring (sum over linear sublayers) → pruning → sign-based merging. The pruning score is clearly defined.
- **Operationally clear merging.** “Top-p% by IGIA then sign-consistent addition into the preceding kept layer” is straightforward to implement and shown with a framework figure.
- **Broad task coverage with efficiency reporting.** Ablations include number of merged layers and sparsity-rate sweeps for the proposed method.

**Weaknesses:**

### (A) Insufficient theoretical grounding for **merging** (most important)
- **Self-inconsistency between pruning and merging.** The paper emphasizes that **layers differ in importance** and uses **IGIA** to make importance-aware pruning decisions. However, during **merging**, contributions from pruned layers are **added with equal weight** whenever signs match—**without any sensitivity weighting** (e.g., IGIA- or Fisher-based) for either donor or receiver layers. This disconnect undermines the rationale that sensitivity should matter.
- **Cross-layer addition lacks justification.** There is no theoretical argument that **elementwise addition across different Transformer blocks** preserves function or yields bounded error, even after sparsification. A first-order approximation, Hessian/Fisher weighting, or Lipschitz-based stability discussion is missing.
- **Actionable request.** Provide a clear **merging objective** (e.g., minimize a first-order loss surrogate), an **error bound**, or at least **IGIA-weighted** or **Fisher-weighted** merges. Otherwise, the method uses importance for pruning but not for merging.

### (B) Comparisons are narrow in **pruning ratio** and **model scale**
- **Single-ratio comparisons.** The main table compares methods **only at 40% sparsity**. While internal ablations vary sparsity for GradPruner, there is **no multi-ratio cross-method** comparison to show whether the advantage holds when pruning is milder or more aggressive.
- **Limited model scaling.** Results center on **7–8B** models (plus a 3B FT reference). It is unclear whether gains **transfer down** to ~1–2B or **up** to ~14B+ models.
- **Actionable request.** Report **accuracy–sparsity curves** for **multiple baselines**, and include **smaller (≈1.7B)** and **larger (≈14B)** models.

### (C) Training recipe & domain generalization
- The domain-specific fine-tuning often appears to train and test within the **same dataset**. This setup makes it hard to assess **out-of-distribution** robustness within the domain.
- **Actionable request.** Include **cross-dataset** evaluations per domain (e.g., train on one medical QA dataset, test on another; for math-style settings: train on MetaMathQA-40k, evaluate on GSM8K / GSM-Plus) to demonstrate generalization beyond the training distribution.

### (D) Minor Issues / Clarity
- **Equation (2) ambiguity/typo.** The equation seems to multiply **the B-gradient twice**, which is likely a typo and inconsistent with LoRA structure. Please clarify the intended mapping and dimensional compatibility.
- **Citation formatting.** Around **line ~310**, the first paragraph’s citation is not enclosed in parentheses, unlike others; please standardize the style.
- **Small typos.** A few truncated terms (e.g., “IGIA-Matri”) and punctuation/spacing glitches around the merging equation.

**Questions:**

please see weakness above.

---

> ### Author Response · Authors · 2025-11-17
> **Response to Reviewer RvRE [1/2]**
>
> We sincerely thank you for taking the time to review our paper and for providing valuable comments and suggestions.
> \subsection{response to weaknesses}
> ## Question: Provide a clear merging objective (e.g., minimize a first-order loss surrogate), an error bound, or at least IGIA-weighted or Fisher-weighted merges. Otherwise, the method uses importance for pruning but not for merging.
>
> In our response to all reviewers, we explained the "Theory of Merging in GradPruner," and further, we theoretically unified Fisher-weighted merging with our method, demonstrating the complementarity of the two approaches. Please refer to the response at the top for detailed theoretical explanations.
>
>
> ## Question: Report accuracy–sparsity curves for multiple baselines, and include smaller (≈1.7B) and larger (≈14B) models.
>
>
> **Response to Report accuracy–sparsity curves for multiple baselines.**  To comprehensively evaluate the relative advantages of GradPruner across different sparsity levels, we have supplemented our original experiments at 40\% sparsity with additional comparisons against baseline methods at 30\% and 20\% sparsity (results shown in the table below).
> We did not consider sparsity levels higher than 40\% because, under such settings, most methods—including GradPruner—exhibit significant performance degradation. Existing baseline methods already struggle to maintain reasonable accuracy at 40\% sparsity, and increasing sparsity further would exacerbate this performance drop, rendering comparisons at higher sparsity levels uninformative.
>
> Table: We conducted comparative experiments between our method and other approaches under different sparsity levels, using the Llama3.1-8B model with full fine-tuning on the HellaSwag dataset. "Dense Model" denotes the accuracy of the unpruned model after fine-tuning on the downstream task.
>
> | sparsity ratio | Dense Model | LLMPruner | Laco | MINITRON | SAT | GradPruner (ours) |
> |---------|-------------|-----------|------|----------|-----|-------------------|
> | 30%| 0.943 | 0.916 | 0.923 | 0.923 | 0.934 | 0.940 |
> | 20%| 0.943 | 0.927 | 0.936 | 0.929 | 0.940 | 0.942 |
>
> The experimental results show the following:
> At 30\% sparsity, GradPruner continues to maintain a clear advantage over all baseline methods.
> At 20\% sparsity (i.e., under milder pruning), some methods—such as Laco and SAT—also demonstrate strong performance; however, our method remains competitive and exhibits consistently robust overall performance.
> These findings indicate that GradPruner’s advantages are not limited to moderate sparsity levels (e.g., 40\%) but also extend to lighter pruning settings, demonstrating its effectiveness and robustness across a range of sparsity regimes.
>
>
> **Response to include smaller (≈1.7B) and larger (≈14B) models**.
> To verify the applicability of GradPruner across different model scales, we have supplemented experiments on both smaller (approximately 1B) and larger (approximately 13B) models, using Llama-3.2-1B and Llama-2-13B as testbeds, respectively. The corresponding results are summarized in the table below.
>
> Table: We evaluate GradPruner on models of different sizes: for Llama-3.2-1B (originally 16 layers), we prune 6 layers; for Llama-2-13B (originally 40 layers), we prune 16 layers. "Dense Model" denotes the accuracy of the unpruned model after fine-tuning on the downstream task.
>
> | Datasets | MedMCQA | FinGPT | HellaSwag |
> |----------|---------|--------|-----------|
> | Llama-3.2-1B (Dense Model) | 0.434 | 0.543 | 0.584 |
> | Llama-3.2-1B (GradPruner) | 0.434 | 0.538 | 0.585 |
> | Llama-2-13B (Dense Model) | 0.542 | 0.806 | 0.907 |
> | Llama-2-13B (GradPruner) | 0.532 | 0.798 | 0.894 |
>
>
> The experiments demonstrate that our method performs effectively across both model scales: it not only maintains strong performance on the larger Llama-2-13B but also exhibits even more pronounced advantages on the smaller Llama-3.2-1B. This indicates that GradPruner’s pruning mechanism is highly scalable, with its effectiveness consistently extending from 1B-scale models to those exceeding 10B parameters, showcasing strong generality.
> The experiments demonstrate that our method performs effectively across both model scales: it not only maintains strong performance on the larger Llama-2-13B but also exhibits even more pronounced advantages on the smaller Llama-3.2-1B. This indicates that GradPruner’s pruning mechanism is highly scalable, with its effectiveness consistently extending from 1B-scale models to those exceeding 10B parameters, showcasing strong generality.

---

> > ### Author Response · Authors · 2025-11-17
> > **Response to Reviewer RvRE [2/2]**
> >
> > ## Question: Include cross-dataset evaluations per domain to demonstrate generalization beyond the training distribution.
> >
> >
> > To verify the generalization ability of the method in cross-dataset scenarios, we supplemented it with cross-dataset evaluation experiments within the domain: pruning and fine-tuning were performed using the original medical domain training set MedMCQA from the paper, and testing was conducted on the medical subset clinical knowledge in MMLU. The relevant results are shown in the attached table.
> >
> > Table: Cross-dataset evaluations in MedMCQA, based on Llama 3.1-8B (Full Fine-tuning). Training data is MedMCQA, and test data is clinical knowledge. The Dense Model represents the accuracy of the unpruned model after fine-tuning on downstream tasks.
> >
> > | Methods        | Dense Model | LLMPruner | Laco | MINITRON | SAT | GradPruner (ours) |
> > |----------------|-------------|-----------|------|----------|-----|-------------------|
> > |                | 0.735       | 0.688     | 0.711| 0.703    | 0.709| 0.717            |
> >
> >
> > Experiments show that, despite a slight performance drop compared to Dense models in cross-dataset settings (i.e., training and testing distributions are not perfectly consistent), GradPruner still maintains a significant accuracy advantage over other existing pruning methods.
> >
> >
> >
> >
> > ## Question: Minor Issues / Clarity.
> >
> > Thank you for the reviewer's detailed and valuable corrections. Regarding formula (2), there is indeed a typo: the gradient of $B$ is incorrectly repeated twice in the original formula. The correct expression should be $\nabla_{W} L(x,y)\_i\overset{\text{sim}}{=}\nabla_{W_B} L(x,y)\_i \cdot \nabla_{W_A} L(x,y)\_i, i \in \\{1,...,t\\}$. We will correct this formula in the paper. In addition, we have noticed that the citation format around line 310 does not use parentheses, which is inconsistent with the style of other citations in the whole text; at the same time, on line 256, "IGIA-Matrix" is misspelled as "IGIA-Matri", and there are punctuation and space issues near the merge equations. These oversights will be corrected uniformly in the final version. Thank you again to the reviewers for their careful review!

---

### Author Response · Authors · 2025-11-17
**To all reviewers:  Theory of Merging in GradPruner**

In principle, **GradPruner is theoretically equal to the isotropic merging method with adaptive importance weights** . Furthermore, **this adaptive importance weighting scheme can also be similarly applied to Fisher merging**. We will elaborate on these two parts in detail below.

### Isotropic merging with adaptive importance weights
To merge $M$layers, isotropic merging with per-layer weights [1] approximates the posterior distribution of each layer using an isotropic Gaussian whose mean is set to the layer’s parameters. This approach introduces layer-specific scalar hyperparameters $\lambda_i$, for $i \in \\{1,\ldots,M \\}$, which can be formally expressed as $\theta^* = \arg\max_{\theta} \sum_{i=1}^M \lambda_i \log p(\theta \mid \theta_i, I),$
where $p(\theta \mid \theta_i, I)$denotes the probability density function of the isotropic Gaussian posterior, and the hyperparameters satisfy $\lambda_i \ge 0$and $\sum_{i=1}^M \lambda_i = 1$. These hyperparameters govern the relative importance assigned to each layer during the merging process. For instance, if all layers are assumed to contribute equally, one may set $\lambda_i = \frac{1}{M}$for all $i$. **However, this approach suffers from two primary limitations**:  (1) How should the importance weights be adaptively determined when the layers exhibit differing levels of importance?  (2) When individual weights within the same layer possess varying degrees of importance, how can adaptive weight-specific coefficients be assigned?

**To address these challenges, our GradPruner introduces an adaptive weighting scheme.** Specifically, consider two layers to be merged: let $W_r$denote the base layer to be retained, and $W_m$the layer to be merged. To assign appropriate weights $\lambda_r^{j}$and $\lambda_m^{j}$to the $j$-th parameter entries $W_r^{j}$ and $W_m^{j}$, **the algorithmic procedure of GradPruner can be formally characterized by the following adaptive weight assignment function**:

$ (\lambda_r^{j}, \lambda_m^{j})= (0.5, 0.5),  if(\frac{\partial L}{\partial W_m^j})^2 \ge T \text{ and } \text{Sign}(W_m^j)=\text{Sign}(W_r^j), $

$ (\lambda_r^{j}, \lambda_m^{j})=(1.0, 0.0), \text{otherwise}$

where $T$ denotes a pruning threshold. This formulation implies that if the squared gradient magnitude of a weight in the merged layer exceeds the threshold $T$ and its sign aligns with that of the corresponding weight in the retained layer, both weights are deemed equally important and are assigned equal coefficients. Otherwise, the weight from the merged layer is considered negligible, and only the retained weight is preserved.

**GradPruner’s importance criterion simultaneously leverages (1) magnitude**—quantified by the squared gradient—**and (2) directional consistency**—captured by the agreement in sign between the two weights—as complementary indicators of parameter significance. This design integrates insights from prior work: methods such as [1,4] emphasize the utility of gradient magnitude in assessing parameter importance, while approaches like [2,3] demonstrate the efficacy of sign consistency as a relevance signal. GradPruner thus unifies both perspectives into a single adaptive framework.
Consequently, the merged weight under isotropic merging with adaptive importance weights is given by  $\theta^j = \lambda_r^j \theta_r^j + \lambda_m^j \theta_m^j.$

### Integrating GradPruner’s Adaptive Weights with Fisher Merging
As outlined above, GradPruner fundamentally constitutes an adaptive strategy for determining importance weights. In this work, we apply this strategy to isotropic merging. Moreover, we demonstrate that GradPruner’s adaptive weighting mechanism can be seamlessly extended to Fisher Merging (FM) [1].

Similar to isotropic merging, FM also formulates the merging objective as
$\theta^* = \arg\max_{\theta} \sum_{i=1}^M \lambda_i \log p(\theta \mid \theta_i, I),$but differs in that it employs the Laplace approximation to the posterior $p$, yielding a Gaussian approximation $\mathcal{N}(\theta, H^{-1})$, where $H$is the Hessian matrix. To render computation tractable, FM approximates the Hessian using the diagonal of the Fisher information matrix $F$. The resulting closed-form expression for the merged parameter is
$\theta^j = \frac{\sum_{i=1}^M \lambda_i F_i^j \theta_i^j}{\sum_{i=1}^M \lambda_i F_i^j}.$

When integrating GradPruner with FM, the adaptive weighting scheme modifies only the assignment of $\lambda_i$, leaving the computation of the Fisher matrix unchanged. Therefore, in the case of merging two layers, the final merged weight becomes
$\theta^j = \frac{ {\lambda_r^j F_r^j \theta_r^j + \lambda_m^j F_m^j \theta_m^j}}{{\lambda_r^j F_r^j  + \lambda_m^j F_m^j }}$ .

### Reference

[1] Merging layers with Fisher-Weighted Averaging

[2] NegMerge: Sign-Consensual Weight Merging for Machine Unlearning

[3] TIES-Merging: Resolving Interference When Merging Models

[4] Model Merging by Uncertainty-Based Gradient Matching

---

### Meta-Review · Area_Chair_jXEa · 2026-01-07

**Summary:**

This paper proposes GradPruner, a gradient-guided structured pruning framework designed to enhance both fine-tuning and inference efficiency for large language models (LLMs). Unlike most existing structured pruning methods that rely on calibration data or additional distillation steps, GradPruner leverages initial gradient information obtained during the first few LoRA fine-tuning steps to estimate parameter and layer importance.

The key innovation lies in computing an Initial Gradient Information Accumulation Matrix (IGIA-Matrix) to quantify layer importance early in training. Based on this metric, GradPruner performs layer-level pruning followed by a layer merging step that sparsifies and merges pruned layers into adjacent retained layers while resolving sign conflicts to reduce destructive interference.

Empirically, GradPruner achieves a 40% reduction in parameters with only 0.99% loss in downstream accuracy, tested on two LLMs (LLaMA3.1-8B and Mistral-7B) across eight diverse benchmarks spanning medical, financial, and general reasoning tasks. It shows consistent performance improvements over strong structured pruning baselines (LLM-Pruner, LaCo, SAT, APT, and MINITRON), and reduces both training and inference time/memory by over 35%.

**Reviewer Concerns:**

All reviewers, especially Reviewer RvRE,  concern the theory of of Merging in GradPruner.
The authors provide a detailed feedback which seems correct to me.
Reviewer RvRE gave an negative score. I checked the rebuttal and found that the authors have solved all concerns of the reviewer and provided more experiments.

**Reviewer Scores:**

NO

---

### Decision · Program_Chairs · 2026-01-26

Accept (Poster)